

# New water-soluble, toxic tracers of wood burning identified in fine brown carbon aerosol using a non-target approach

Vinh Nguyen[1], Bartłomiej Witkowski[1*], Tomasz Gierczak[1]

[1]University of Warsaw, Faculty of Chemistry, al. Żwirki i Wigury 101, 02-089 Warsaw, Poland

*Correspondence to:* Bartłomiej Witkowski (bwitk@chem.uw.edu.pl)

**Abstract.** The molecular composition of water-soluble fine ($PM_3$) brown carbon aerosol ($BrC_{aq}$) generated by combustion of wood was studied with ultra-performance liquid chromatography coupled with electrospray ionization time-of-flight mass spectrometry (UPLC-ESI-ToF/MS) using a non-target analysis (NTA) workflow. The NTA analysis workflow based on MS-DIAL and MS-FINDER showed the best performance of the five software tested. Structures of 361 out of the 420 water-

soluble organics in BrC were tentatively identified for the first time. The total emission of fine, water-soluble $BrC_{aq}$ was approx. 1 g per kg of wood burned, comparable with the emission factors of some semi-volatile organics from open biomass burning. Potential precursors of aqueous secondary organic aerosols ($_{aq}$SOAs) and toxic molecules were selected among the newly identified molecules.

The newly identified harmful tracers of fine BrC included plant and wood care products, alkaloids, and fungal metabolites.

Fungal metabolites were also identified among the potential precursors of $_{aq}$SOAs with high Henry's law constants values, alongside natural compounds occurring in roots and leaves, diterpenoids, flavonoids, anthraquinones, and coumarins. The release of these natural and man-made compounds is possible during wildfires and domestic uses of biomass. The atmospheric lifetimes calculated for the newly identified precursors of $_{aq}$SOAs showed that natural dyes, bacterial and fungal metabolites, and (aromatic) glucosides can undergo aqueous OH oxidation in cloud water. Such molecules can produce low-volatility

products without decomposing due to their large carbon backbones. Many new potential chromophores were also identified in BrC, including natural dyes and molecules with conjugated double bonds and aromatic rings.



## 1 Introduction

Biomass is organic material from plants and animals and often refers to non-fossil fuels, like wood, pellets, and straw. Biomass is used around the globe for generating heat and energy, including domestic and industrial uses (Antar et al., 2021; Tomlin,
2021). Over 3 billion people use solid (bio)fuels for cooking and heating, which globally accounts for 41% of households (Amegah and Jaakkola, 2016). Open biomass burning (BB) is also widespread in some regions due to cultural and economic practices (Admasie et al., 2018). Furthermore, the use of biomass aims to limit the reliance of the energy industry on nonrenewable fossil fuels and to decrease the net emissions of greenhouse gasses (GHG) (Tomlin, 2021). At the same time, the large-scale use of biomass as a renewable, carbon-neutral fuel in power and heating plants remains debatable (Stubenrauch
and Garske, 2023).

Open BB includes burning crop residues and (Hartner et al., 2024b), vegetation fires intensified by man-made climate change (Jones et al., 2024). Around the globe, more frequent and widespread wildfires pose a serious threat to humans, infrastructure, and ecosystems. In the 2023-2024 fire season, the global area burnt is estimated at $3.9 \times 10^6$ km$^2$, causing many fatalities and billions of dollars in damages in the US, Canada, the EU, and Asia (Jones et al., 2024).

In addition to direct exposure effects, open BB emits large amounts of pollutants, which can affect areas far away from the source (Jones et al., 2024; Laskin et al., 2025). BB is a major global source of methane (5-15%), CO (30-50%), NO$_x$ (20%), anthropogenic CO$_2$ (18%), and the second largest source of non-methane volatile organic compounds (VOCs) (Pan et al., 2020). Furthermore, BB is the major source of black carbon (BC) and primary organic aerosols (POAs), accounting for 57±2% and 87±2% of the global emissions, respectively (Andreae, 2019; Jiang et al., 2024). BC and organic aerosols (OAs), including
also light absorbing OAs, the so-called brown carbon (BrC) emitted by BB account for up to 70% of the total emission of fine particulate matter (PM$_{2.5}$) into the atmosphere (Jiang et al., 2024; Yadav and Devi, 2019).

BB pollutants, including GHGs and fine PM, affect the air quality, and climate and harm human health (Jiang et al., 2024). All fine PMs influence the hydrological cycle by initiating the formation of clouds and ice crystals (Bellouin et al., 2020). Fine aerosols, including biogenic secondary organic aerosols (BSOAs) (Tsigaridis and Kanakidou, 2018), also scatter light, thereby
reducing the amount of radiation reaching the Earth's surface (Kahn et al., 2023). Unlike some BSOAs, BC and BrC absorb the incoming solar radiation, exhibiting positive radiative forcing, which increases global temperature (Laskin et al., 2025).

Due to its high atmospheric abundance, BC is the second largest contributor to man-made radiative forcing (Matsui et al., 2018). The contribution of BrC to direct radiative forcing of BC is likely substantial but the estimates vary between 20 and 70% (Laskin et al., 2025). BC absorbs light from infrared (IR) down to the ultraviolet (UV) region whereas BrC exhibits a
much narrower (wavelength-dependent) absorption in the UV-Vis region of the electromagnetic spectrum (Saleh, 2020; Laskin et al., 2025).

The radiative forcing of BrC estimated by global models varies from 0.03 to 0.57 W m$^{-2}$ (Li et al., 2023). This uncertainty is, in part, due to incomplete data about the sources, formation mechanisms, atmospheric transformations, light absorbance, and chemical composition of BrC (Li et al., 2023). For this reason, identifying the pollutants emitted from combustion sources has



been the subject of extensive research (Li et al., 2023; Young et al., 2021). Thousands of unique molecules contribute to BrC aerosols, including saccharides, halogenated, nitrated, and halogenated phenols, polycyclic aromatic hydrocarbons, terpenoids, resin acids, dioxins, alkanes, oxygenated and aromatic and organosulfur compounds, etc. (Divisekara et al., 2023; Young et al., 2021; Brege et al., 2021). At the same time, most of these compounds remain unidentified, because a comprehensive, molecular-level characterization of BrC presents a considerable analytical challenge (Brege et al., 2021; Divisekara et al.,

2023; Laskin et al., 2025).

Furthermore, the chemical composition of BrC is not only forbiddingly complex but also evolves during transport and chemical processing (the so-called chemical aging) in the atmosphere (Moise et al., 2015; Laskin et al., 2025). Formation and evolution of BrC in the atmosphere involve various gas, aqueous, and multiphase reactions, such as chemical aging by UV radiation and inorganic radicals (Zhao et al., 2015). Particularly, the currently poorly characterized (photo)chemical processing of BB

emissions in atmospheric hydrometeors largely influences the composition, light absorption, and toxicological and physicochemical properties of BrC (Wong et al., 2019; Choudhary et al., 2023). For instance, the aqueous oxidation of BrC by OH initially enhanced the UV-Vis absorbance but led to bleaching following prolonged exposure (Lei et al., 2025; Hems et al., 2020). However, due to the limited data on the water-soluble organics compounds (WSOCs) emitted by BB, our understanding of the climate and health effects of BrC remains incomplete (Li et al., 2023).

This work aimed to study the WSOCs in fine BrC (BrC$_{aq}$) emitted by the pyrolysis of woody biomass, focusing on toxic molecules and those with high Henry's law constants (H, M/atm) values. The latter group can readily dissolve in atmospheric hydrometeors and undergo further chemical processing, resulting in (light-absorbing) aqueous secondary organic aerosols ($_{aq}$SOAs) (Lei et al., 2025; Go et al., 2024). $_{aq}$SOAs are likely important but poorly characterized compounds of fine atmospheric PM (Su et al., 2020).

In the work presented, to gain detailed, molecular-level insights into the composition of BrC$_{aq}$, analyses were carried out with ultra-performance liquid chromatography coupled with electrospray ionization time-of-flight mass spectrometry (UPLC-ESI-ToF/MS) using a non-target analysis (NTA) workflow. NTA identifies (annotates) unknown compounds based on the HR-MS data using databases and *in-silico* fragmentation prediction (Hulleman et al., 2023; Vosough et al., 2024). Therefore, NTA is a promising approach for resolving the molecular complexity of BrC (Laskin et al., 2025). This work presents the first

application to studying BrC$_{aq}$ generated by wood pyrolysis (Divisekara et al., 2023; Young et al., 2021; Hartner et al., 2024b). The immense amounts of data acquired with the modern LC/MS and GC/MS instruments require advanced data-processing algorithms, involving deconvolution, feature detection, and annotation (Hohrenk et al., 2020a). Over the past decade, there has been a considerable increase in the availability of software for HR-MS data mining (Hohrenk et al., 2020a). However, the number of features extracted and identified in NTA largely depends on the data processing workflow (Hohrenk et al.,

2020a; Wartmann et al., 2024). Several NTA workflows were tested in this work, including competitive Fragmentation Modeling for Metabolite Identification (CFM-ID) (Wang et al., 2021), Metaboanalyst (Pang et al., 2024), Global Natural Products Social Molecular Networking (GNPS) (Aron et al., 2020), MS-DIAL, MS-FINDER (Tsugawa et al., 2015b), and





MZmine (Schmid et al., 2023), using a mixture of model BB pollutants. Afterward, the best-performing workflow was used to analyze $BrC_{aq}$.

Here, BrC was generated in the $N_2$ atmosphere in a new, custom-designed combustor to simulate the pyrolysis of wood during vegetation fires (Gao et al., 2024). The composition of $BrC_{aq}$ was compared with the previously published data and, >360 tracers of wood pyrolysis were tentatively identified for the first time. Several WSOCs in $BrC_{aq}$ with high H values and toxicities ($LD_{50}$) were also first identified in this work.

Additionally, quantitative analysis was performed with LC/MS using surrogate standards (STs) to identify the major
components of $BrC_{aq}$ (Pieke et al., 2017). Quantitative results obtained using STs, total organic carbon (TOC) analyzer, and gravimetric method showed that estimating relative ionization efficiencies for WSOCs in $BrC_{aq}$ yielded reasonably accurate results.

## 2. Experimental section

Materials and reagents are listed in Section S1 in the Supplementary Information (SI).

### 2.1 Biomass combustor and fine particle collection

A diagram of the newly constructed biomass combustor is shown in Fig. 1

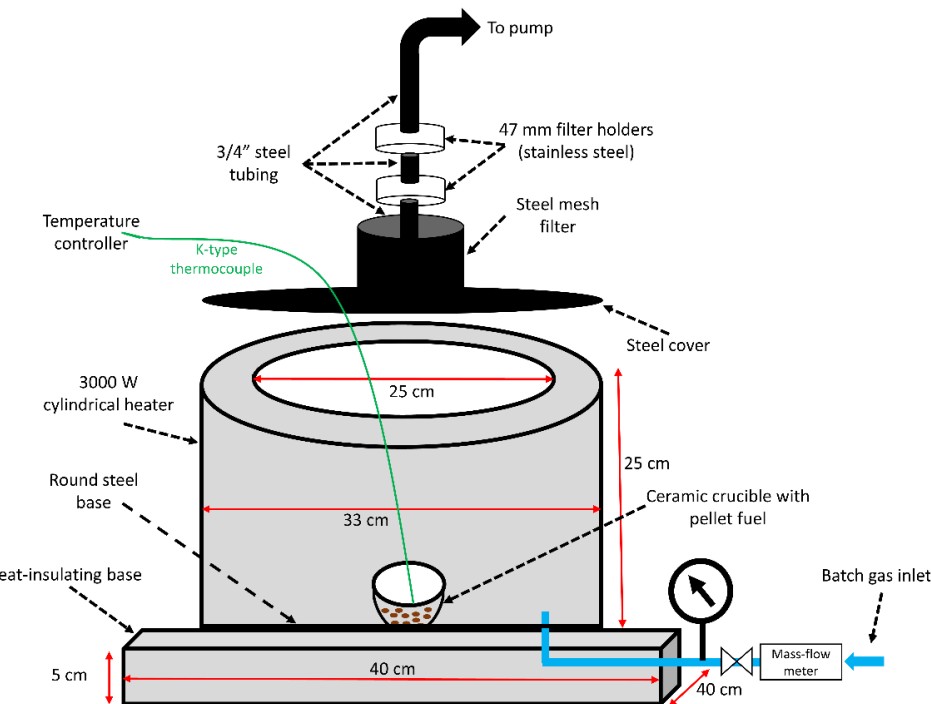

**Figure 1: Diagram of the biomass combustor**



The combustor consisted of a 3 kW cylindrical resistance heater with a round steel base that was placed on top of a heat-
insulating board – Fig. 1. The heater was secured to the base with heatproof cement, and surrounded by a 5-cm thick layer of
fireproof insulation. The outer wall of the combustor was a steel cylinder with an additional layer of a heat-reflecting mat. The
combustor base was equipped with the batch gas inlet, which ran inside the insulating board – Fig. 1. A controller was used to
adjust the combustion temperature within ± 5°C. Pellets were placed in a ceramic crucible with a K-type thermocouple
connected to the temperature controller and inserted in the middle of the fuel stack. The flow of batch gas and the pumping
speed were adjusted with manual valves, and the pressure inside the combustor was 1 atm.

Mixed wood pellets (25 g) were heated at 350°C for 2 hours and the bath gas flow ($N_2$) was maintained at 1.5 L/min. The
stream from the combustor cooled down to approx. 80°C before reaching the sampling assembly. First, the emissions passed
through a stainless steel mesh filter (mesh size 0.5 mm  - 50 μm, Fig. 1) to remove coarse particles. Afterward, two 47 mm
filters were used; the first filter (hydrophobic PTFE, pore size 3 μm, Fluoropore, FSLW07400) was followed by a second
PTFE-coated glass microfiber filter (EMFAB TX40H120-WW, Pallflex, aerosol retention 99.95%). During the sampling, both
filter assemblies were kept at 80°C using a heating sleeve (10 cm I.D., not included in Fig. 1) to avoid condensation of water
and other semi-volatiles.

Before sampling, filters were placed in a desiccator with dry molecular sieves for 24 hours. Particles collected on the second
filter ($PM_3$) were extracted using 2 mL of water via mechanical agitation, and filtered through a 0.22 μm PTFE filter before
analysis.

**2.2 Liquid chromatography coupled to mass spectrometry**

Analyses were performed using the Q-TOF LCMS-9030 system (Shimadzu) using an Acquity HSS-T3 column (Waters, 100
mm × 2.1 mm, 1.8 μm). The mobile phase consisted of 0.1% formic acid in water (A) and 0.1% formic acid in ACN (B); the
injection volume was 2 μL. The mobile phase flow rate was 0.25 mL/min, and the column temperature was 30°C. The
following gradient elution program was used: initially, 5% for 10 minutes, then linear increase to 25% over 5–22 minutes; held
at 25% for 13 minutes, then linear increase to 95% over 35-39 minutes, held at 95% for 5 minutes, then linear decrease to 5%
over 1 minute, and held at 5% for 5 minutes. The total analysis time was 50 minutes.

The mass spectrometer was equipped with an electrospray ion (ESI) source, operating in positive or negative ionization modes.
The nebulizing, drying, and heating gas flow were set at 3.0 L/min, 10 L/min, and 10 L/min, respectively. The interface
temperature was 300°C. The TOF mass resolving power was approx. 45 000.  Spectra were collected in the data-dependent
acquisition (DDA) model; 100 – 600 m/z (precursor selection, even time 1 s), 40-600 m/z (product ion scan, even time 0.1 s),
number of dependent events 3, intensity threshold 3000, delay time 1 s, charge state 1. Each sample was analyzed six times,
at positive and negative ionization modes utilizing three collision energy (CE, V) values: 0-7 V for (low), 10±5 V (medium),
and 17±5 V (high), and the CE spread was 5 V.



### 2.3 Mass spectrometric data processing workflows

The raw data files were first processed with MS-DIAL or MZmine (Fig. 2). Afterward, the workflows based on MS-FINDER, CFM-ID, Metaboanalyst, GNPS and, MZmine were tested using 59 organic molecules, representing common BB pollutants, including derivatives of cinnamic acid, nitrophenols, and polycarboxylic, furoic and fatty acids (Section S1). The final workflow, used for analyzing $BrC_{aq}$ was based on MS-DIAL and MS-FINDER.

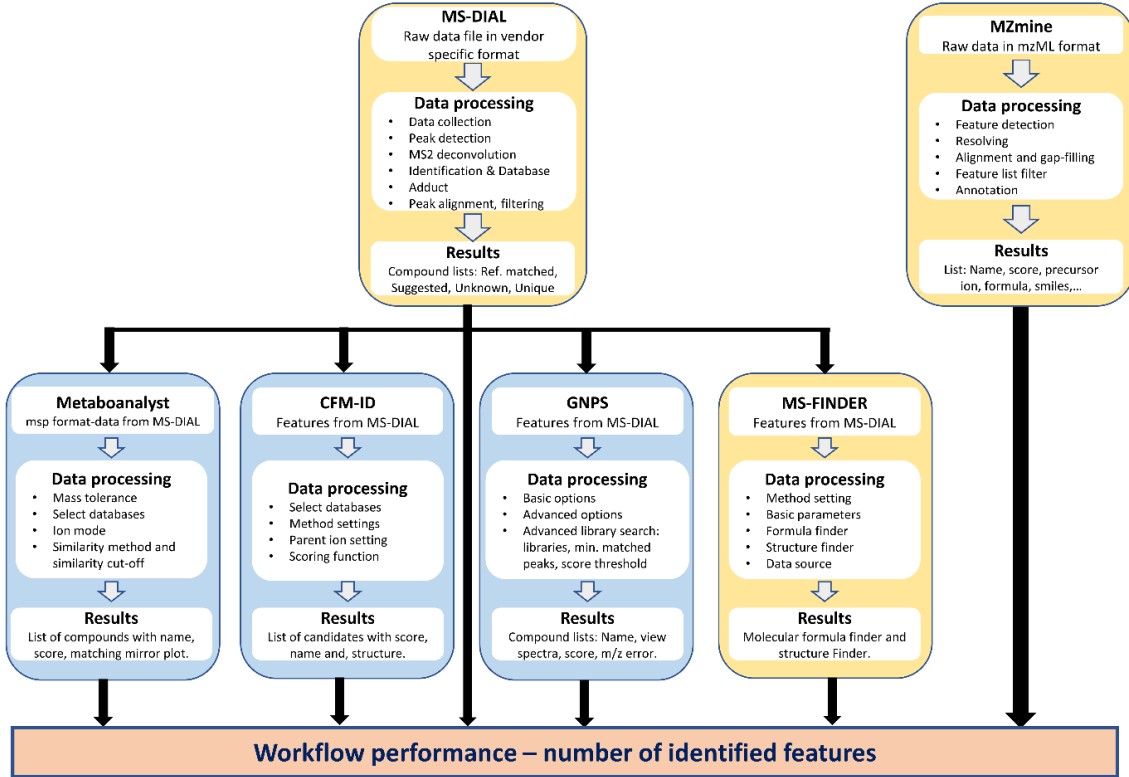

**Figure 2. The MS data processing software was tested in this work. All data processing was performed with a Dell Vostro 3020T desktop PC equipped with an Intel Core i7-13700 CPU, 1TB SSD drive, and 64 GB of 3200MHz DDR4 RAM.**

The same MS databases, including internal and used uploaded libraries, were used in workflows based on MS-DIAL, MS-FINDER, and MZmine; GNPS allows user-defined databases while the other web-based platforms (MetaboAnalyst and CFM-ID) are limited pre-selected MS libraries – Table S2 (Lai et al., 2018; Vaniya et al., 2017; Mona, 2024; Ms-Dial, 2024; Tsugawa et al., 2015b). Workflows based on MZmine, GNPS, MetaboAnalyst, and CFM-ID are described in Section S2.1.

### 2.3.1 MS-DIAL

MS-DIAL (v5.3.240617) was used for the raw data processing, spectra deconvolution, and feature extraction and annotation – Fig. 3.



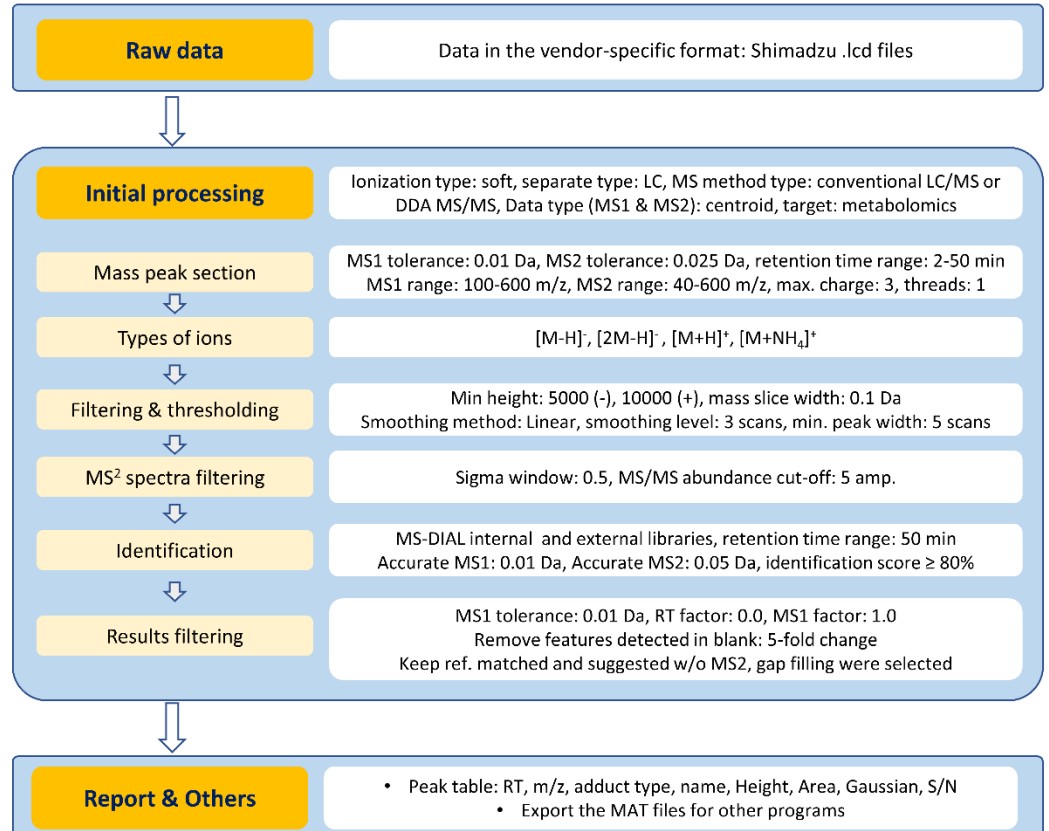


**Figure 3. MS-DIAL workflow**

The identification criteria included retention time, precursor *m/z*, isotopic ratio, and MS/MS spectra (Fig. 3). MS/MS spectra were indispensable for feature annotation and distinguishing isomers. Here, the compound with the highest total score above 80% was assigned to each focus peak (Tsugawa et al., 2015a). In cases where DDA lacked MS/MS spectra, the MS/MS

similarity value was zero, and the structure was proposed based on the MS1 score.

The Microsoft Access Table (MAT) file, which included both MS and MS/MS spectra exported from MS-DIAL was used as input data for MS-FINDER, CFM-ID, GNPS, and MetaboAnalyst but not MZmine (Fig. 1). Features not annotated by MS-DIAL were further analyzed using MS-FINDER (Section 2.3.2).

**2.3.2 MS-FINDER**

MS-FINDER v3.61 (Fig. 4) is a highly versatile tool for formula predictions, fragment annotations, and structure elucidations. In addition to the internal libraries embedded in the MS-FINDER use an in-silico MS/MS spectra predictor. Assigned structures were ranked based on their scoring.



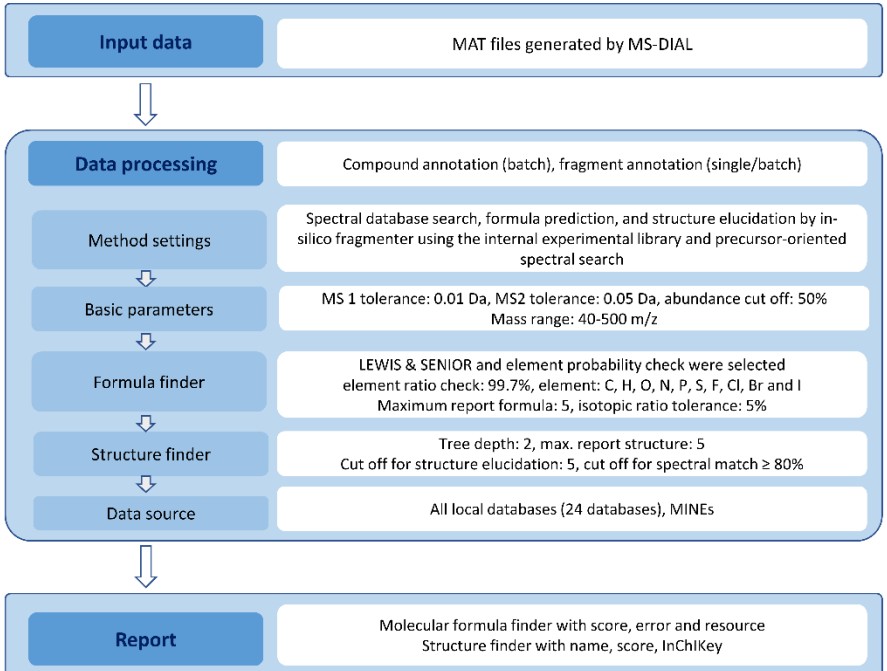

**Figure 4. MS-FINDER workflow**

In MS-FINDER, the formula candidates were considered based on mass error, isotopic ratio, product ions annotation, neutral-loss ions annotation, and database score (Fig. 4). Structural candidates were ranked using a weighted scoring system, integrating bond dissociation energies, mass similarities, fragment linkages, and, most importantly, nine hydrogen rearrangement rules during bond cleavages. The final molecular structure rankings were determined using a combined formula and structure scores (Tsugawa et al., 2016; Blaženović et al., 2018)

**2.4 Confidence levels in annotation**

Confidence levels in NTA are showcased in Table 1 (Schymanski et al., 2014).

**Table 1**. Confidence levels in the NTA workflow

| Confidence level | Required data | Classification | Description | Example |
|---|---|---|---|---|
| 1 | MS spectra, MS/MS spectra, library MS/MS spectra, and retention time data matching | *Confirmed* | Input spectra matched with reference spectra by MS-DIAL with a retention time confirmation | |



| | | | | |
|---|---|---|---|---|
| 2 | MS spectra, MS/MS spectra, library MS/MS spectra matching | *Reference matched* | Input spectra matched with reference spectra by MS-DIAL | |
| 3 | MS, MS/MS, experimental data | *Suggested structure* | From MS-DIAL, and MS-FINDER with MS/MS spectra | |
| 4 | MS isotope/adduct | *Suggested formula* | From MS-DIAL with MS spectra | $C_7H_6O_3$ |
| 5 | MS | *Unknowns* | With MS spectra and an unknown compound with MS/MS spectra | *m/z* 137.0228 |

For the features for which a confidence level ≥4 was obtained (elemental formula assignment), the van Krevelen diagram, Kendrick mass defects, double-bound equivalents (DBE), and average carbon oxidation states ($OS_c$) were calculated – Section S3 (Kroll et al., 2011).

**2.5 Semi-qualitative analyses with surrogate standards**

Surrogate standards (STs) were used to quantify organic compounds detected in filter extracts; this approach assumes that closely eluting compounds have similar response factors (RFs) (Pieke et al., 2017). Five STs were used in each ionization mode, with retention times between 2 and 40 min (Table S1). Working standards were prepared in (MeOH/H$_2$O, 1:1, v/v) with concentrations from 0.001 to 0.05 mg/L.

RF for STs were calculated with eq. I.

$$RF_{ST} = \frac{A_{ST}}{C_{ST}} \qquad \text{(I)}$$

In eq. I, $RF_{ST}$ represents the ratio between the peak area $A_{ST}$ and the concentration ($C_{ST}$, mg/L) of the ST. The analyte concentrations were calculated using eq. II (Pieke et al., 2017).

$$C_{unknown}\left(\frac{mg}{L}\right) = \frac{A_{unknown}}{RF_{closest\ ST}} \qquad \text{(II)}$$

In eq. II, $RF_{closest\ ST}$ is the response factor of ST closest in the retention time to the analyte, and $A_{unknown}$ is the chromatographic peak area of the unknown compound. $C_{unknown}$ is the concentration of the analyte in mg/L. In eq. II, the total concentration of WSOCs in the filter extracts was calculated as a sum of concentrations of the individual analytes. Due to the



use of STs, an uncertainty of 50% for all concentrations obtained with eq. II was assumed (Kruve, 2019; Evans et al., 2024a;
Malm et al., 2021; Pieke et al., 2017).

The total ($\Sigma_{LCMS}$) and individual amounts of WSOCs emitted were derived with eq. III.

$$\Sigma_{LCMS}(\tfrac{g}{kg}) = \frac{\sum C_{unknown} \times DF \times V_{total}}{\Delta m_{fuel}} \qquad \text{(III)}$$

In eq. III, $\Sigma C_{unknown}$ is the sum of analytes (mg/L) from eq. II,  DF is the dilution factor, $V_{total}$ is the total volume of extraction
solvent (2 mL), $\Delta m_{fuel}$ is the decrease in the mass of pellets (g) corrected for the water content (8.0%).

**2.6 Total organic carbon analysis**

Analyses were conducted using a Shimadzu TOC-5050A analyzer with an ASI-5000A autosampler. The instrument was
calibrated with the standard solution of 4-nitrophenol and glucose between 4 and 40 (mgC/L). A squared linear coefficient of
determination ($R^2$) of 0.9996 was obtained, and the RF values for the two standards were practically identical.

Samples were diluted with water to a ca. 20 mgC/L and filtered through a 0.22 µm PTFE syringe filter. Before injection, 50 µL
of 2M HCl was added to each sample, followed by sparging with $O_2$ for 2 minutes. The injection volume was 21 µL and each
sample was injected three times, which yielded a precision <1%.

$$\Sigma_{TOC}(\tfrac{g}{kg}) = \frac{C_{TOC} \times DF \times V_{total}}{\Delta m_{fuel}} \times \frac{1}{0.625} \qquad \text{(IV)}$$

In eq. IV, $C_{TOC}$ is the measured concentration of TOC (mgC/L), DF is the dilution factor, $V_{total}$ is the total volume of extraction
solvent (2 mL), $\Delta m_{fuel}$ is the fuel mass (g) corrected for the water content (8.0%) and, $0.625 \pm 0.144$ ($2\sigma$) is the factor used to
convert mgC/L units to mg/L. This conversion factor was the average carbon content of the analytes detected with LC-ToF/MS
and the main source of the uncertainty of the TOC measurements.

**2.7 Gravimetric measurements**

Aqueous samples (0.4 mL) were filtered through a 0.22 µm PTFE filter and evaporated to dryness at 30°C using a centrifugal
vacuum evaporator (Labconco, model no. 7810033). Afterward, the dried residues were placed in a desiccator for 24 hours to
remove the leftover moisture. This procedure yielded ≥1 mg of residue, which was weighed using an analytical microbalance
(Radwag, model no.XA52.5Y).

The amount of BrC$_{aq}$ emitted was calculated with eq. V.

$$TC_{grav}(\tfrac{g}{kg}) = \frac{(m_{residue} - m_{blank}) * V_{total}}{V_{use} * \Delta m_{fuel}} \qquad \text{(V)}$$

In eq. V, $m_{residue}$ is the mass of the residue after evaporation (mg), $m_{blank}$ is the mass of the blank (mg), $V_{total}$ is the total
volume of extraction solvent (2 mL), $V_{use}$ is the volume of extract used for evaporation (0.4 mL), and $\Delta m_{fuel}$ is the fuel
mass (g) corrected for the water content (8.0%).



**2.8 Henry's law constants and toxicity estimates**

Acute toxicity, measured as $LD_{50}$ (mg/kg) and H values (M/atm), were estimated using the VEGA-QSAR regression model (KNN, v. 1.0.0) and Henry's law model (OPERA, v.1.0.1) (Mansouri et al., 2021; Mansouri et al., 2018).

Eq. VI and VII were used to derive the total toxicity and H scores for individual compounds in $BrC_{aq}$.

$$H\ score\ (arb) = H \times emission\ factor\ (LC/MS) \qquad (VI)$$

$$LD_{50}\ score\ (arb) = \frac{1}{LD_{50}} \times emission\ factor\ (LC/MS) \qquad (VII)$$

Using eq. VI and VII, the candidates for $_{aq}$SOAs precursors and the most harmful molecules were identified by considering their properties and amounts emitted.

**2.9 Quality and control measurements**

Blank filters were prepared without the fuel (Section 2.1). The extract from the blank filter was then used to correct the TOC and LC-MS analysis results. After each experiment, the combustor was cleaned by heating the chamber to 550ºC and flushing with air (5.0 L/min) for 1 hour to oxidize the residues. The sampling assembly was sonicated in detergent, rinsed with distilled water, and organic solvent, and dried in the oven.

**3. Results and discussion**

**3.1 Non-target workflow selection**

Workflows (Sections 2.3 and S2.1) were evaluated using model compounds  – Fig. 5.





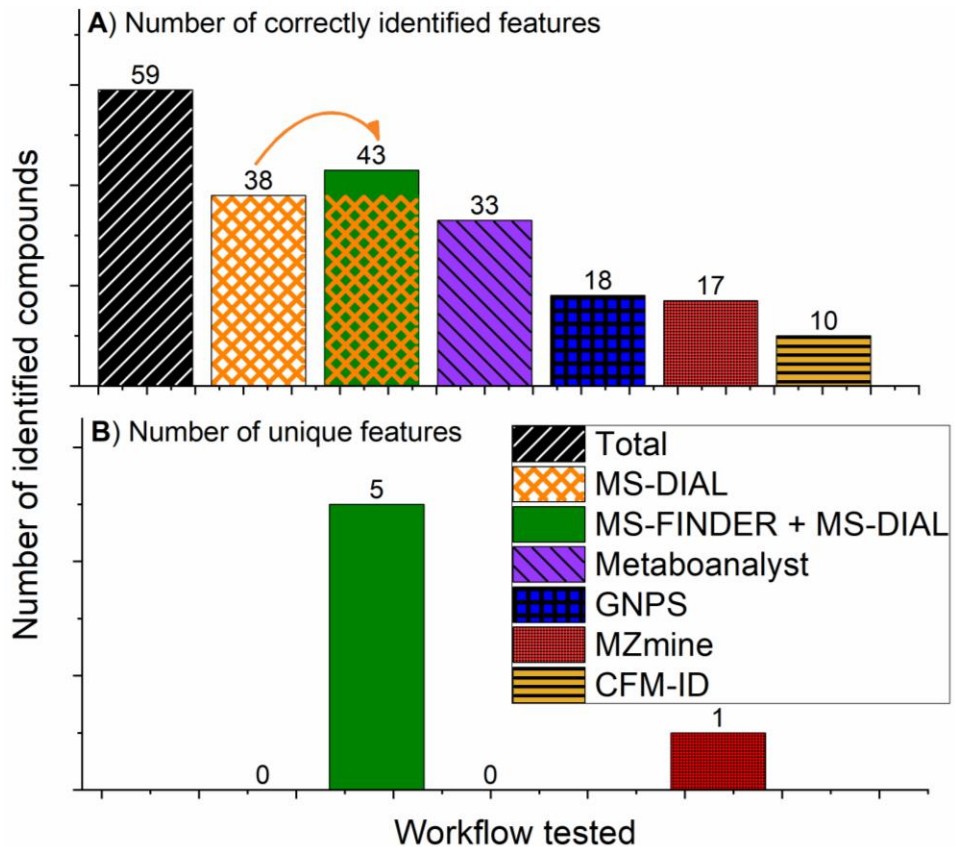

**Figure 5. Workflow performance evaluation with the model compounds; number of correctly identified compounds**
**(A) and number of compounds identified only by a given software (B). The data presented are provided in Appendix 1.**

The optimal workflow, combining MS-DIAL and MS-FINDER, correctly identified 73% of standards (Fig. 2), similar to earlier studies (Young et al., 2021; Black et al., 2021). The combination of MS-DIAL and MS-FINDER also identified 5 unique features not annotated by any other workflow, whereas MZmine identified only one unique analyte – Fig. 5.

In addition to searching MS databases, MS-FINDER identifies unknown compounds using *silico* fragmentation algorithms 240 (Blaženović et al., 2018). The performance of the combined workflow (Fig. 5) is attributed to the accurate prediction (simulation) of the fragmentation spectra in MS-FINDER (Tsugawa et al., 2016; Su et al., 2023). While MS-DIAL excelled in reproducibility and peak picking, MS-FINDER demonstrated a robust capability to identify unknown BB pollutants (Wartmann et al., 2024; Mallmann et al., 2023).

Despite using different peak-picking parameters (Fig. 3 and Fig. S1), the number of features extracted by MZmine and MS-245 DIAL was very similar. While MZmine extracted more features than MS-DIAL, it is not necessarily an indicator of higher data quality (Rivera-Pérez and Garrido Frenich, 2024). MZmine enhanced the accuracy with noise filter and duplicate peak filter (Heuckeroth et al., 2024b), while MS-DIAL highlighted interference reduction using its "blank filtering" capability





currently absent in MZmine (Heuckeroth et al., 2024a). These functions made MS-DIAL well-suited for NTA, further complemented by the user-friendly interface.

MZmine successfully identified only 29% of standards, despite using the same databases as MS-DIAL and MS-FINDER (Table S2), consistent with earlier findings (Wartmann et al., 2024). Such a result is attributed to differences in mass peaks-picking algorithms, which largely impacted the accuracy of extracted *m/z* values. In MZmine, mass detection was performed via the exact mass algorithm using the full width of half maximum paradigm to determine peak centers and extract *m/z* values and intensities (Hohrenk et al., 2020b). On the other hand, MS-DIAL employed peak detection algorithms rooted in a linearly

weighted smoothing average, regarding retention time and accurate mass (Tsugawa et al., 2014). This process was based on differential calculus principles and noise estimations, forming the backbone of MS-DIAL peak detection methodology (Tsugawa et al., 2015a).

All web-based platforms, MetaboAnalyst, GNPS, and CFM-ID utilize measured and predicted spectra in their matching functions. MetaboAnalyst correctly identified 33 standards while GNPS and CFM-ID, identified only 18 and 10 out of 59

standards, respectively (Fig. 5).

In most MS data mining programs, MS/MS fragments are transformed into vectors. The cosine value, representing the angle between the input and reference vectors from databases, is used to quantify their similarity (Li et al., 2021b). The MetaboAnalyst platform integrated similarity scores in a single value, derived neither from the vector direction (dot-product) nor the vector magnitude (entropy) (Pang et al., 2024). In MS-DIAL, the similarity score is based on a combination of dot-

product (also used in MetaboAnalyst) and secondary input spectra. The secondary input spectra remove fragments that did not appear in the reference spectra of the candidate compound. It decreases the impact of unwanted fragments derived from isotopic and background noise (Tsugawa et al., 2015a). Consequently, for the selected standards (Section S1), the accuracy of results generated by MS-DIAL and MetaboAnalyst is higher than other tested programs (Fig. 5) (Tsugawa et al., 2015a).

Databases embedded in CFM-ID contain nearly nine million spectra, but most are simulated (Table S2) (Databases, 2024).

Since CFM-ID identified only 10 standards (Fig. 5), the accuracy of these prediction algorithms was somewhat insufficient (Bremer et al., 2022). Another drawback of the CFM-ID is the requirement to provide three fragmentation spectra for each unknown compound, acquired at low, medium, and high CE to enhance the identification accuracy, which greatly prolongs the analysis time (Chao et al., 2020). Nevertheless, simulating MS/MS spectra with machine learning is a promising approach to identifying "known unknowns" via *in-silico* fragmentation and "unknown unknowns" in non-target identification (Bremer

et al., 2022; Russo et al., 2024)

## 3.2 General characteristics of fine BrC$_{aq}$

The combined workflow (MS-DIAL and MS-FINDER) exhibited the best performance and was used to analyze the composition of BrC$_{aq}$. Analysis of BrC$_{aq}$ yielded 4086 features; this number was reduced to 2121 features with identification level ≥4 (elemental formula assignment, see - Table 1). These features were categorized based on their elemental composition

- Fig. 6





**Figure 6. The van Krevelen Diagram (A), Kendrick Mass Defect (B), DBE vs (C+N) atoms (C), and OS$_c$ vs C atoms (D)**

**plots for unique molecular compounds of water-soluble particulate matter identified by MS-DIAL and MS-FINDER. The combination of symbols C, H, O, N, and S refers to the molecules composed of the listed elements. The size of the circles corresponds to the relative concentrations in the aqueous extracts obtained with LC/MS using STs (Section 2.5).**



The van Krevelen diagram (Fig. 6A) revealed the clustering of O/C and H/C ratios between 0.2-0.4 and 0.7-1.3, respectively, corresponding to molecules derived from ligins and tannins (D'andrilli et al., 2015; Hartner et al., 2024b), which were the most

abundantin $BrC_{aq}$, both number and concentration-wise - Fig. 6 (Laszakovits and Mackay, 2021; Moschos et al., 2024a). Similar results were previously reported for the ambient and chamber-generated BB aerosols (Fleming et al., 2018; Evans et al., 2024b). Furthermore, a Kendrick Mass Defect plot (Fig. 6B) revealed a homolog series of CHON, CHO, and CHONS molecules with MWs between 150 and 300 Da, similar to OAs emitted by dung and brushwood burning (Fleming et al., 2018). Highly oxidized $C_xH_yO_{3-6}$ compounds with $C_xH_yO_4$ formulas were the most frequently detected, constituting 73% of 96

annotated features detected with (-ESI). On the other hand, nitrogen-containing organic compounds, such as $C_xH_yNO_{2-4}$, and $C_xH_yN_z$, were predominantly observed in positive ion mode, accounting for approximately 38% of all detected compounds - Fig. 6 (Ma et al., 2024; Li et al., 2024).

Double-bond equivalent (DBE) values (Fig. 6C), corresponding to the degree of unsaturation, ranged from 4 to 6 for lignin pyrolysis products to approximately 7−8 for coumarins, 10−12 for stilbenes, and flavonoids (Moschos et al., 2024a; Koch and

Dittmar, 2006). In Fig. 6C, 60% of (-ESI) and 40% of (+ESI) candidates fell within the potential BrC chromophores region, positioned between the lines for conjugated polyenes and, linear fullerene-like hydrocarbons (Tang et al., 2020; Lin et al., 2018; Siemens et al., 2022; Moschos et al., 2024a; Sun et al., 2024). The chromophore candidates were primarily lignin-like CHO molecules followed by nitrogen-containing CHON compounds identified, previously identified in wood-burning BrC (Fleming et al., 2018; Evans et al., 2024b).

The $BrC_{aq}$ exhibited the carbon oxidation state ($OS_c$) values ranging from -2 to 1 (Fig. 6D), consistent with observations in organic aerosols within Earth's atmosphere (Moschos et al., 2024a; Kroll et al., 2011). Furthermore, for approximately 83 % of the annotated compounds, the $OS_c$ was between -2 and 0, characteristics of unaged emissions (Nihill et al., 2023). Hence, the general characteristics and oxidation state of $BrC_{aq}$ corresponded to the unaged OAs emitted by BB (Moschos et al., 2024a; Fleming et al., 2018; Smith et al., 2009; Song et al., 2018; Li et al., 2024).

**3.3. Quantitative and qualitative analyses of fine $BrC_{aq}$**

Quantitative analyses of the water-extractable fraction of fine BrC were performed with LC-MS using STs (Section 2.5), TOC (Section 2.6), and gravimetric analysis (Section 2.7) – Fig. 7.





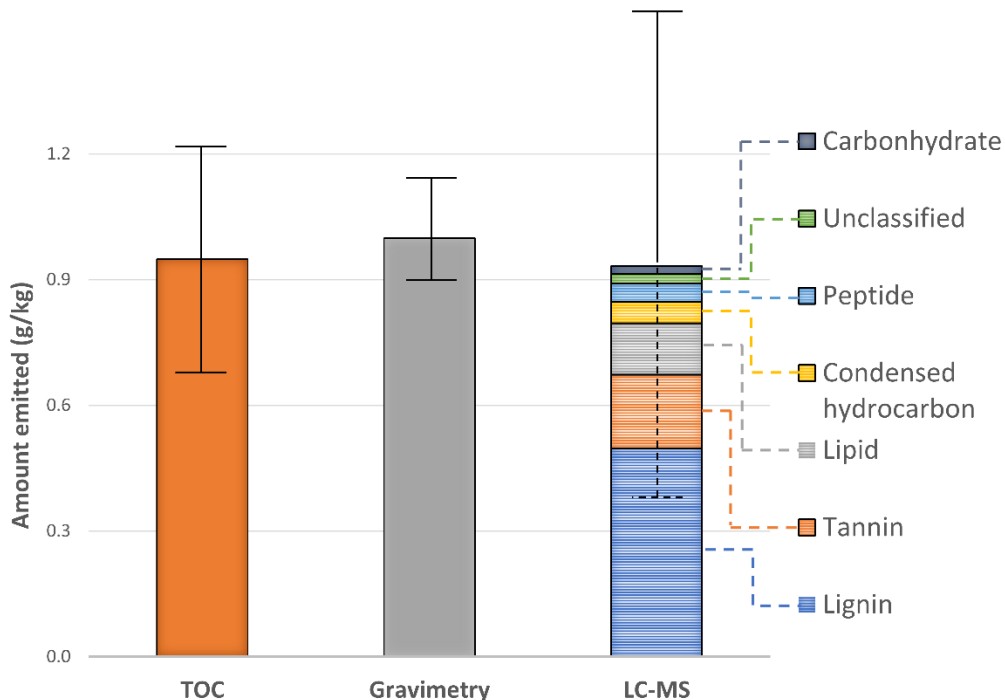

**Figure 7. The total concentration of BrCaq (PM3) generated by pyrolysis of wood pellets at 350°C measured with LC-MS ($\Sigma_{LCMS}$) and TOC ($\Sigma_{TOC}$) and gravimetry ($\Sigma_{grav}$). The data presented are provided in Table S6.**

The WSOCs detected in BrC$_{aq}$ were classified as fatty and carboxylic acids, alkanes and aliphatic hydrocarbons, aromatic compounds, peptides, and polypeptides (Fig. 7) based on their general characteristics derived from elemental composition – Fig. 6 (Smith et al., 2009; Merel, 2023; Zherebker et al., 2024). BrC$_{aq}$ was dominated by pyrolysis products of lignin and tannins, followed by lipids, peptides, hydrocarbons, condensed hydrocarbons, and carbohydrates (Seo et al., 2020; Shahid et al., 2019; Divisekara et al., 2023; Smith et al., 2020; Noblet et al., 2024; Hartner et al., 2024c; Kawamoto, 2017).

An excellent agreement between the $\Sigma_{LCMS}$, $\Sigma_{TOC}$ and, $\Sigma_{grav}$ was obtained but, the first two values are largely implied by uncertainties - Fig. 7. Moreover, the values of $\Sigma_{TOC}$ and, $\Sigma_{grav}$ may be affected by the loss of some volatile organics during sparging with $O_2$ (Section 2.6) and drying under vacuum (Section 2.7) and, such molecules were shown to contribute to BrC (Sinha et al., 2023; Priestley et al., 2024). Furthermore, the value of $\Sigma_{grav}$ may include fine soot (not removed by the 0.22 μm filter, Section 2.7) and water-soluble inorganics, which can contribute 8.9-21% to fine BrC (Yadav et al., 2023; Trubetskaya, 2022). Nevertheless, the results (Fig. 7) indicate that the semi-qualitative analysis with LC/MS (Section 2.5) reasonably estimated the total concentration of BrC$_{aq}$ (Seo et al., 2020; Shahid et al., 2019; Divisekara et al., 2023; Smith et al., 2020; Noblet et al., 2024; Hartner et al., 2024c; Evans et al., 2024a).

The total emission of polar, water-soluble organics from wood pyrolysis at 350°C, representing the (oxygen-depleted) smoldering conditions, was ca. 1 g/kg of fuel burned (Fig. 8). The values derived for individual WSOCs (mg/kg) derived with eq. III (Section 2.5) pertains solely to the water-soluble fraction of fine BrC and, should be regarded as a rough estimate of the





actual emission factors (EFs) (Pokhrel et al., 2021). Nevertheless, the amounts of individual WSOCs emitted (Tables S7 and S8) are comparable with the EFs (mg/kg) range reported for the minor BB pollutants, including semi-volatile organics (Akagi et al., 2011), indicating a significant contribution of water-soluble organics to BB emissions.

**3.4 Molecular composition and properties of the fine, water-soluble brown carbon**

The BrC$_{aq}$ sampled from the combustor was analyzed in DDA mode and the detected features were annotated by MS-DIAL and MS-FINDER – Fig. 8

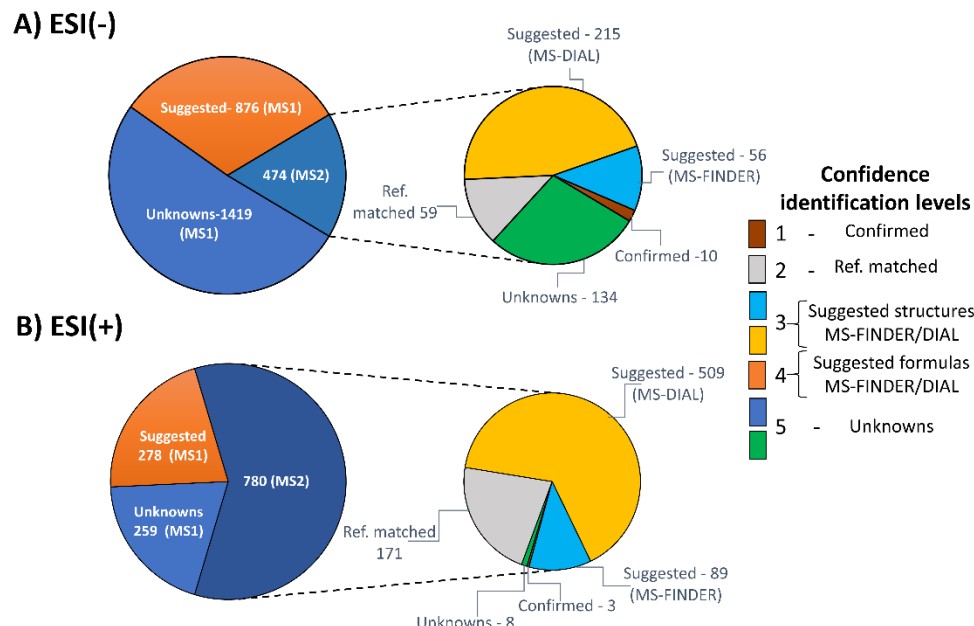

**Figure 8. The number of features extracted and identified by MS-DIAL and MS-FINDER in fine BrC generated by**
**pyrolysis of woody biomass at 350°C; negative (A) and positive (B) ionization modes. All compounds identified in both ionization modes are listed in Tables S4 and S5.**

The features extracted were first annotated based on the MS1 spectra at level 4 confidence level (Table 1). At this stage, 51% (-ESI) and 20% (+ESI) of features were denoted as unknowns – Fig. 8. Subsequently, MS2 spectra were further analyzed, and 12% (-ESI) and 59% (+ESI) of total features were annotated at confidence level >3. Furthermore, confidence level >4 was
obtained for 44% and 80% of the detected features in negative and positive ionization modes, respectively (Yang et al., 2023; Schymanski et al., 2014). MS databases for positive ion mode contained over 326,000 records compared to 53,337 records in negative mode (Table S2), which likely enhanced the annotation results for (+ESI) – Fig. 8 (Ng, 2021). Hence, the database plays a critical role in the structural elucidation of NTA but, currently, the number of (freely available) MS/MS databases focused on environmental pollutants is limited (Ng, 2021).

Major WSOCs detected in BrC$_{aq}$ were tentatively identified using NTA – Fig. 9.







**Figure 9. Tentatively assigned structures of the five most abundant WSOCs in BrCaq classified as derivatives and pyrolysis products of lignins (A), tannins (B), lipids (C), condensed aromatics and hydrocarbons (D), peptides (E), and carbohydrates (F). Unlabeled areas correspond to unidentified molecules in each group. All structural assignments via NTA and identification confidence levels are provided in Tables S4 and S5.**

The major tracers of BB identified include phenols and methoxyphenols originating from lignin and tannin pyrolysis (Fig. 9A-B) (Li et al., 2021a; Hartner et al., 2024a; Wan et al., 2019). As previously reported, products derived from aromatic alcohols sinapyl, coniferyl, and p-coumaryl, constituted 53.4% of WSOCs generated by the combustion of lignin (Fleming et al., 2018; Kawamoto, 2017; Simoneit, 2002). Furthermore, mono and poly-carboxylic acids including succinic, glutaric, adipic, sorbic, and azelaic acids (Hu and Yu, 2013; Narukawa et al., 1999), were identified as major compounds in lipid fraction (Fig. 9C), consistent with the polar SVOCs from laboratory-generated BB emissions and fine particles collected at an urban location (Sengupta et al., 2020; Hu and Yu, 2013; Shen et al., 2022). Condensed aromatics and hydrocarbons, peptides, and carbohydrates were minor components of $BrC_{aq}$ (Figs. 9D-F). While some studies reported higher emissions of these molecules, their contribution varies considerably depending on the pyrolysis temperature (Zhang et al., 2024; Oros et al., 2006; Chang et al., 2024).

Structures of 361 out of the 420 WSOCs detected in $BrC_{aq}$ were tentatively identified for the first time (Tables S4 and S5). Of the 50 most abundant molecules in $BrC_{aq}$ (Table S7) identified at level ≥3, 28% was previously detected in ambient and laboratory-generated (light-absorbing) OAs (Yee et al., 2013; Sengupta et al., 2020; Divisekara, 2023; Moschos et al., 2024a; Fleming et al., 2020; Oros et al., 2006; Graham et al., 2002; Hartner et al., 2024a; Bianco et al., 2016; Oros and Simoneit, 2001; Chan et al., 2020).

These (newly) identified structures were used to estimate H and $LD_{50}$ values (Section 2.8) listed in Tables S8 and S9. The estimated properties exhibited almost no correlation with the elemental composition and the general characteristics of WSOCs identified at the level ≥4, including the number of O, N, C, and S atoms and N/O ratio for each molecule (Table S10). Statistically significant (p value<0.05), but weak (r= 0.561) correlation was observed between H values and the number of oxygen atoms. Hence, the properties of polar (complex) molecules in $BrC_{aq}$ cannot be even roughly constrained using simple parameters derived from HR-MS measurements, and more sophisticated models, such as quantitative structure-activity relationships (QSARs), are necessary (Mansouri et al., 2021; Mansouri et al., 2018).

**3.5 Potential precursors of $_{aq}$SOAs in fine $BrC_{aq}$**

Because some structural assignments were ambiguous, which is inevitable in NTA (Hulleman et al., 2023; Hohrenk et al., 2020a), the list of potential $_{aq}$SOAs precursors with the 50 highest H scores (Table S8) was further refined - Fig. 10.



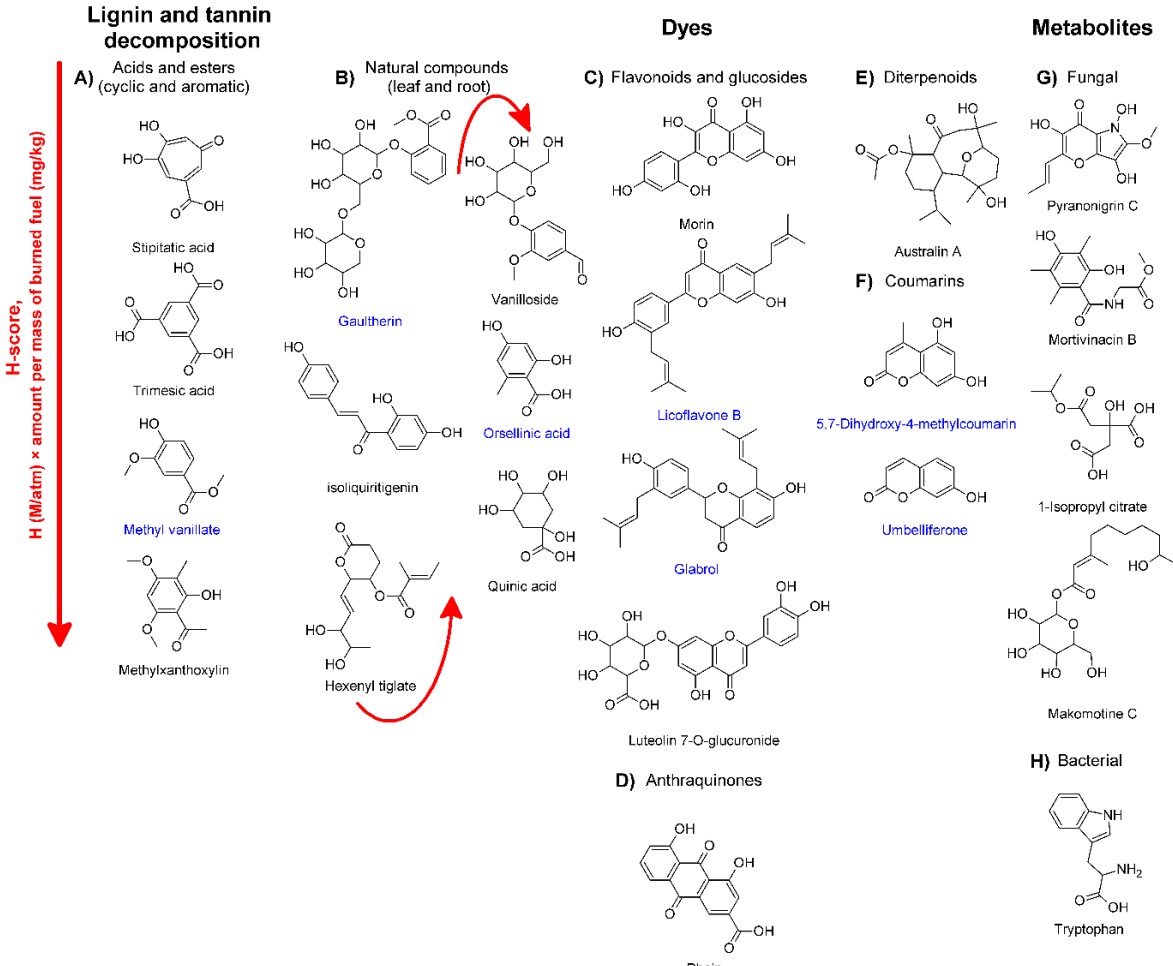

**Figure 10. Newly identified potential precursors of $_{aq}$SOAs with the highest H scores (eq. VI in Section 2.8). The structures shown were assigned at levels 2 and 3, with names shown in black and blue, respectively. Identification levels in NTA are showcased in Table 1, lower is better. Seven out of twenty-two compounds shown were identified at the 2nd**

**level, the highest possible confidence without using authentic standards.**

The newly identified $_{aq}$SOAs precursor candidates included cyclic and aromatic acids and esters, which can be classified as typical BB tracers (Fig. 10A) (Laskin et al., 2025; Wan et al., 2019). Furthermore, relatively large quantities of natural compounds present in roots and leaves (Fig. 10B), and natural dyes, including flavonoids (aglycones and glucosides, Fig. 10C) and anthraquinones (Fig. 10D) and coumarins (Fig. 10E) were also identified (Lin et al., 2016; Moschos et al., 2024b; Laskin

et al., 2025; Huang et al., 2022). In addition to high solubility in water, natural dyes substantially contribute to the BrC absorption between 300 and 370 nm (Laskin et al., 2025; Zhou et al., 2021). Moreover, one diterpenoid (Fig. 10E) and several fungal and bacterial metabolites (Figs. 10G and H) were detected (Laskin et al., 2025; Wei et al., 2019).



Molecules shown in Fig. 10 (or their analogs) can be obtained from commercial suppliers, and serve as model precursors for investigating the combustion-related $_{aq}$SOAs. To date, such studies have focused primarily on (nitrated) phenols (Lei et al.,

2025; Jiang et al., 2021; Hems and Abbatt, 2018; Witkowski et al., 2022). Hence, studying these newly identified WSOCs can shed new light on the formation and evolution of light-absorbing OAs in the atmospheric hydrometeors, even if they only serve as proxies of ambient BrC (Hems et al., 2020; Liu et al., 2020; Laskin et al., 2025). The atmospheric lifetimes of the compounds shown in Fig. 10 due to the reaction with OH (the major daytime atmospheric oxidant) are further discussed in Section 4.

### 3.6 Toxic compounds identified in fine BrC$_{aq}$

The newly identified molecules with the highest LD$_{50}$ scores (Table S9) are shown in Fig. 11

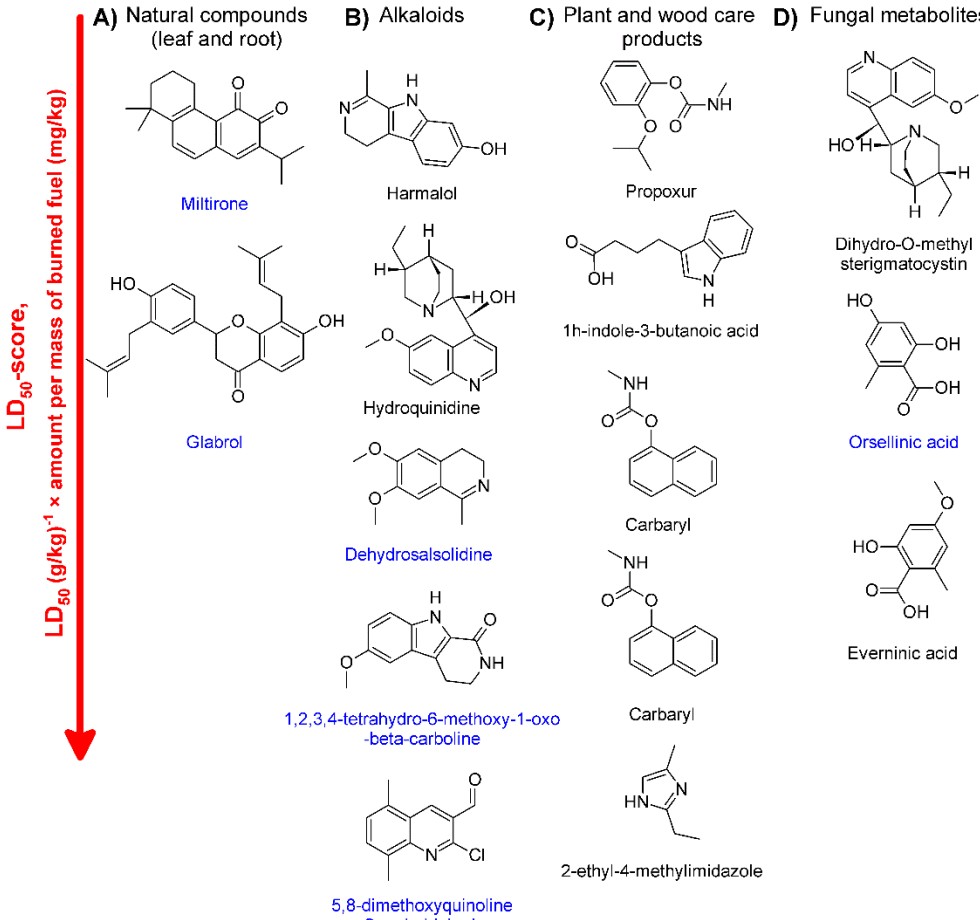

**Figure 11. Newly identified, harmful components of BrC$_{aq}$ with the highest LD$_{50}$ scores (eq. VII in Section 2.8). The structures shown were assigned at levels 2 and 3, with names shown in black and blue, respectively. Identification levels in NTA are showcased in Table 1, lower is better. Six out of fifteen compounds shown were identified at the 2$^{nd}$ level,**

**the highest possible confidence without using authentic standards.**



Of the 50 WSOCs with the highest $LD_{50}$ scores, 32 were classified as moderately ($50<LD_{50}<500$) toxic, and harmful ($500<LD_{50}<2000$) with five highly toxic ($LD_{50} <50$) molecules – Table S9 (Gadaleta et al., 2019). Several N-containing compounds were identified among the toxic components of $BrC_{aq}$ (Fig. 11) (Pflieger and Kroflič, 2017; Majewska et al., 2021). However, no correlation between the $LD_{50}$ values and the number of N atoms or N/O ratios was observed (Table S10), again
underscoring that such general classifications can be unreliable (Young et al., 2021; Khan et al., 2021). The newly identified WSOCs included natural compounds (Fig. 11A), particularly alkaloids (Fig. 11B), which were seldom identified as (harmful) BB tracers (Young et al., 2021; Nizkorodov et al., 2011).

Furthermore, plant and wood care products (Fig. 11C), and fungal metabolites (Fig. 11D) were detected in a toxic fraction of $BrC_{aq}$ (Laskin et al., 2025; Wei et al., 2019; Růžičková et al., 2021). The release of insecticides from commercial pellet fuel
was previously observed as low-temperature pyrolysis (here 350°C was used – Section 2.1), which is insufficient to decompose such molecules (Růžičková et al., 2021). Forest protection with man-made chemicals and the production of wood pellets from different waste materials contribute to releasing these toxic WSOCs (Růžičková et al., 2021; Cesprini et al., 2021; Alakoski et al., 2016). Furthermore, the raw material for wood pellets usually contains between 70 and 95% tree wood and thus can include other kinds of forest biomass (Cesprini et al., 2021), but secondary contamination during storage (e.g. fungal growth) is also
possible (Alakoski et al., 2016). In vegetation fires, properties cannot be controlled but the current wood pellet quality standards do not consider the organic (micro)pollutants (Cesprini et al., 2021; Alakoski et al., 2016).

### 3.7 Atmospheric implications

The chemical aging of BB aerosols is connected with the formation and evolution (bleaching) of BrC (Wong et al., 2019; Zhao et al., 2015; Choudhary et al., 2023; Witkowski et al., 2022). Therefore, the lifetimes of potential $_{aq}$SOA precursors shown in
Fig.10 due to reaction with OH were estimated with eq. VIII (Sarang et al., 2021).

$$\tau = \frac{1}{\left(\frac{k_{OH_{gas}}}{H_{OH}^{cc}} + k_{OH_{aq}} H_{BrC}^{cc} \omega \right)[OH]_{aq}} \qquad (VIII)$$

In eq. VIII, τ is the overall gas and aqueous phase lifetime due to the reaction with OH. The equilibrium concentrations of individual $BrC_{aq}$ and OH in both phases are derived using the dimensionless $H_{OH}^{cc}$ and $H_{BrC}^{cc}$ values (Sander, 2015), and liquid water content (LWC, ω, unit $m^3/m^3$). The bimolecular reaction rate coefficients ($k_{OH_{gas,}}$ and $k_{OH_{aq}}$ – Table S11) at 298K were
estimated with $py$SiRC model (Sanches-Neto et al., 2021).

The lifetimes estimated with eq. VIII for the potential precursors of $_{aq}$SOAs (Fig. 10) cover a wide range of values from less than one minute to several hours – Fig. 12A.



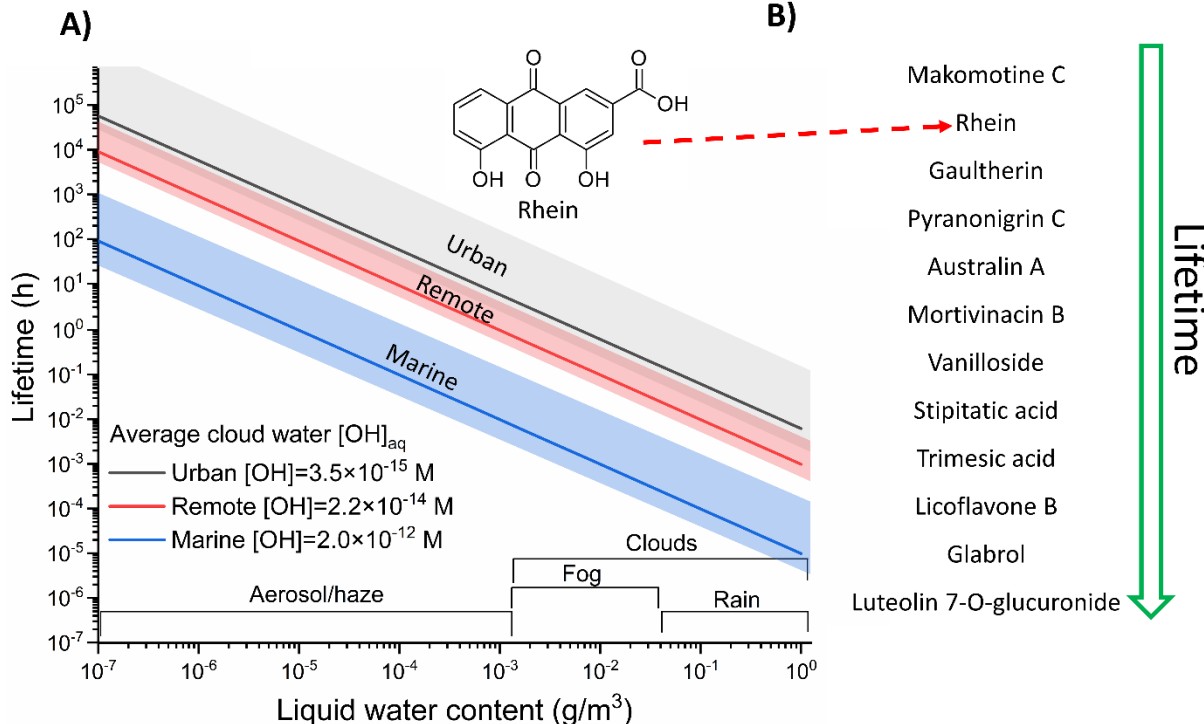

**Figure 12. A - estimated τ values (eq. VIII) for anthraquinone rhein due to the reaction with OH in different atmospheric hydrometeors, shaded areas represent the highest and lowest [OH]$_{aq}$. Only one sample plot is shown because the same profiles were obtained for all compounds listed in Table S8 (molecules with the highest H$_{scores}$). B - the potential precursors of $_{aq}$SOAs with the shortest lifetimes are listed.**

Considering the aqueous processing of BB emissions in the atmosphere, continental (urban and remote) cloud-water processing is the most relevant with marine scenarios being feasible following long-range transport (Che et al., 2022; Laskin et al., 2025). The τ values for all compounds with high H scores (Table S8) were affected by liquid water even in urban clouds, with the lowest [OH]$_{aq}$ (Fig. 12A). An estimated time of air-parcel interaction with the cloud is very long (18h) (Herrmann et al., 2015), but the cloud droplet lifetimes are < 1 min (Kumar et al., 2013; Paulson et al., 2019). The estimated τ values also do not consider aqueous sources of OH, primarily Fenton (like) reactions which can increase [OH]$_{aq}$ by several orders of magnitude (Kuang et al., 2020; Paulson et al., 2019).

Under these assumptions, of all potential precursors (Table S8), the most reactive molecules and those emitted in the highest quantities were selected (Fig. 12B). Thus, these newly identified WSOCs (Fig. 10) will likely undergo aqueous OH oxidation under realistic atmospheric conditions. Analyzing the yields of $_{aq}$SOAs from these precursors would require a separate investigation. However, compounds with large carbon backbones (Fig. 10) will likely yield low-volatility products following a reaction with OH without decomposing, at least during the early stages of oxidation (Fig. 12) (Herraiz and Galisteo, 2015).



## 4. Conclusions


This work revealed new, water-soluble tracers of wood combustion in fine, water-soluble BrC, including insecticides, wood-care products, and bacterial and fungal metabolites. The release of these compounds is possible during vegetation fires and domestic uses of wood and pellet fuel, contributing to the adverse health effects of open and domestic BB. Furthermore, in the case of vegetation fires, even 70% of the fuel is consumed under oxygen-depleted and smoldering conditions (Akagi et al.,

2011), likely favoring the release of polar, water-soluble organics, such as the molecule newly identified in this work (Chen and Bond, 2010). Global, annual emissions from open BB (not including domestic uses) were estimated at ca. 2600 TgC including 18.6 Tg of particle-bound organic carbon (Liu et al., 2024). At the same time, very little data is still available about the EF of the polar, higher-molecular-weight WSOCs emitted by open BB, even though such molecules are released during combustion and pyrolysis without decomposing (Li et al., 2021a). This work revealed that the amount of water-soluble organics

emitted under oxygen-depleted conditions may be comparable with the EFs of non-polar, lower-molecular weight BB tracers. Such molecules can form light-absorbing SOAs following oxidation reactions in different hydrometeors and contribute to the adverse health effects of BB emissions

Studying the light-absorbing properties of $BrC_{aq}$ was beyond the scope of this work, however, new potential chromophores, including natural dyes and molecules with conjugated double bonds and aromatic rings, were also identified (Tables S4-5).

The results presented provide new insights into the structures of BrC chromophores (Laskin et al., 2025)

*Data availability.* The raw data can be obtained by contacting the corresponding author.

*Author contributions.* BW designed the study, developed the methodology, analyzed the data. VN carried out the experiments, optimized the methodology, and processed the raw data. TG supervised the experiments, analyzed the data, . All authors contributed to the interpretation of the results and contributed to manuscript writing and editing.

*Competing interests.* The authors declare that they have no conflicts of interest

*Acknowledgments.* This work was carried out at the Biological and Chemical Research Centre, University of Warsaw, established within the project co-financed by the European Union from the European Regional Development Fund under the



Operational Programme Innovative Economy, 2007–2013. We thank the anonymous reviewers for the very insightful

comments and suggestions.

*Financial support.* This project was funded by the Polish National Science Centre: grant number 2021/43/B/ST10/00931.

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
