# Peer review of "New water-soluble, toxic tracers of wood burning identified in fine brown carbon aerosol using a non-target approach"

_EGUsphere, 2025_

## Author Comment (AC1)

We would like to thank Reviewer for the thoughtful comments and suggestions that helped us improve several aspects of our work, particularly the clarity and robustness of the revised article. Our point-by-point responses to the specific comments provided, together with the list of changed made in the revised manuscript, are provided below.

**Reviewer 1 comments**

**General Comments:**

This manuscript presents a thorough comparison of multiple NTA workflows for analysis of water-soluble brown carbon aerosol samples. This is a much-needed contribution to the field, and additionally, the best workflow was used to identify new potential precursors to aqueous SOA. My scientific comments are minor, and I have a few comments to address clarity and technical issues with the manuscript. Overall, this manuscript is worthy of publication in ACP.

**Specific Comments:**

**Reviewer 1:** It is not clear how model compounds were selected. Were these compounds chosen from studies focused on aqueous brown carbon, or on biomass burning in general? Or perhaps they were chosen because of commercial availability? Since workflow performance was assessed on the ability to correctly identify these compounds, it seems important to communicate how exactly these standards were chosen and cite references that informed the choice, if applicable.

**Author's response:** The model compounds were selected to best mimic the molecular composition of biomass-related, water-soluble BrC, considering the previously published data and availability of standards. Most of the model compounds are aromatics, functionalized (aromatic)acidic, phenols, or N-containing molecules previously detected in laboratory-generated and ambient BB emissions. Some model compounds are used as surrogates for similar molecules.

**Changes made in the manuscript:** Table S1 was added in the SI, with references justifying the use of selected standards as (model)tracers of organic compounds detected in biomass-burning emissions. This table is now referenced at the beginning of Section 2.3: "First, the workflows based on MS-FINDER, CFM-ID, Metaboanalyst, GNPS, and MZmine were tested using 59 model compounds. These molecules were selected to mimic the molecular composition of water-soluble BrC emitted by BB based on previously published results and standards availability (Tables S1 and S2), and included derivatives of cinnamic acid, nitrophenols, and polycarboxylic, furoic, and fatty acids."

**Reviewer 1:** I would welcome the addition of a comment from the authors about how well their combustion apparatus simulates real BB conditions, but do not feel it is absolutely necessary prior to publication. I am always a bit skeptical of these small-scale reactors in terms of their applicability to real conditions, and I am even more skeptical given this is a brand-new combustion system that has not been previously characterized. That being said, I don't think the very nice NTA work done here is invalidated by the choice of combustor.

**Author's response:** We agree that laboratory-scale reactors are never fully representative of the ambient conditions. At the same time, without access to large-scale installations [1, 2], or sampling ambient wildfire emissions, smaller reactors are often used to generate (proxy) BrC mixtures [3, 4]. These (small-scale setups) are generally considered capable of generating (proxy) BB emissions. A recent work also reported using liquid smoke as a surrogate for biomass burning aerosol [5].

In the presented work, we use oxygen-depleted conditions because there is little to no oxygen inside the logs or in the diffusion flame, as well as deep inside the fire [2, 3, 5, 6]. The results of chemical analyses also indicate that our approach generated representative BrC aerosols, in terms of general chemical characteristics (Fig. 6 and discussion in Section 3.2) and specific molecular markers, reported in similar samples (Fig. 9 and Tables S8 and S9).

We also agree that this new setup would benefit from a more detailed characterization, particularly analyses of size distribution, light absorption, and scattering of particles exiting the sampling assembly. However, we believe that in the context of the presented work, such analyses are not of critical importance, as mentioned by the referee.

**Changes made in the manuscript:** The conditions used in the combustor were better justified in Section 1: "Here, BrC was generated in the $N_2$ atmosphere in a new, custom-designed combustor. Wood pyrolysis was conducted to simulate combustion conditions during vegetation fires, as oxygen is absent inside logs and in the deeper zones of the fire [2, 3, 5, 6]."

**Reviewer 1:** I found Figure 9 to be confusing – if the unlabeled areas correspond to unidentified molecules in each group, how are the authors determining that those molecules belong to that group? The information shown in Table S4 and S5 only added to my confusion. It seems like a reference to Tables S6 and S7 here would be more appropriate.

**Author's response:** The compounds are assigned to specific groups based only on the elemental composition, as presented in Fig. 6 in the main text; the corresponding equations are also provided in Section S4 in the SI. Hence, it is possible to assign a given molecule to a group without any structural elucidation, as showcased in Table 1 for confidence level 4. Please note that all annotated compounds are listed in Tables S8 and S9 (formerly S4 and S5), and are sorted from confidence level 1 (confirmed

with authentic standard) to 3, listing all structural assignments, in addition to the sample structures presented in Fig. 9.

**Changes made in the manuscript:** The following lines were added in the caption of Figure 9: "Detected compounds were assigned to these groups based on assigned elemental formulas (confidence level 4 in Table 1) as presented in Fig. 6. Unlabeled areas correspond to unidentified molecules in each group. Only five major components from each group are shown, and all structural assignments via NTA and identification confidence levels 1 to 3 are listed in Tables S8 and S9"

**Technical Corrections:**

**Reviewer 1:** I found Table 1 to be confusing in the sense that it implies no identifications were made at the highest level of confidence (e.g. Level 1). I suggest the authors add an example in the first row, since the SI shows that the method successfully reached Level 1 for a variety of compounds.

**Author's response:** We agree with this comment; matching a given molecule with an authentic standard wasn't emphasized sufficiently in Table 1. The absence of a sample structure can also confuse the readers.

**Changes made in the manuscript:** The description of confidence level 1 in Table 1 was revised as "Confirmed with authentic standard*",* and the sample structure was added in the corresponding row.

**Reviewer 1:** Several times throughout the manuscript, something to the effect of "level ≥ 4" is used. The context of these sentence conflicts with the literal meaning. In other words, what I think the authors are attempting to communicate is 'a level of confidence higher than 4,' which would correspond to a level with a *numerical* value ≤ 4. This occurs at lines 279, 344, 367, and 373 (and perhaps elsewhere). While a reader familiar with the Schymanski confidence level scheme will likely understand what is being implied, I suggest these be reworded to be more clear for those less familiar with NTA.

**Author's response:** We agree with this comment; the use of this notation (level ≥ 4) is incorrect.
**Changes made in the manuscript:** All uses of this notation **"**level ≥ 4" were removed from the text and replaced with **"**features annotated at least level 4 (elemental formula assignment)**"**, **"**confidence level 4, or higher**"** **"**identified at least at level 3 (Table 1)**"** **"**annotated at least at level 4 (Table 1)**"**. Furthermore, the reader is now referred to Table 1 more often throughout the text.

**Reviewer 1 technical comments:**

**All of the technical comments were revised as suggested by the Referee. Please see below for our answers to specific comments.**

Commas in strange places in lines 299 and 301

**Changes made in the manuscript:** This sentence was revised as **"Double-bond equivalent (DBE) values (Fig. 6C), reflecting the degree of unsaturation, ranged from 4–6 for lignin pyrolysis products, 7–8 for coumarins, and 10–12 for stilbenes and flavonoids"**

Line 290: missing a space between the words "abundant" and "in"

Line 293: I suggest spelling out molecular weights and organic aerosols to be more clear.

Remove "identified" before the comma in line 303

Line 307: "characteristics" should be "characteristic"

Line 311: By the time I got here, I had totally forgotten what "STs" were. Consider not using an acronym all to refer to the surrogate standards.

**Changes made in the manuscript:** The use of the abbreviation STs was removed from the main text and the supporting information. The remaining technical corrections were revised according to the Referee's suggestions.

**References**

(1) Stockwell, C. E.; Yokelson, R. J.; Kreidenweis, S. M.; Robinson, A. L.; DeMott, P. J.; Sullivan, R. C.; Reardon, J.; Ryan, K. C.; Griffith, D. W. T.; Stevens, L. Trace gas emissions from combustion of peat, crop residue, domestic biofuels, grasses, and other fuels: configuration and Fourier transform infrared (FTIR) component of the fourth Fire Lab at Missoula Experiment (FLAME-4). *Atmos. Chem. Phys.* **2014**, *14* (18), 9727-9754. DOI: 10.5194/acp-14-9727-2014. Stockwell, C. E.; Veres, P. R.; Williams, J.; Yokelson, R. J. Characterization of biomass burning emissions from cooking fires, peat, crop residue, and other fuels with high-resolution proton-transfer-reaction time-of-flight mass spectrometry. *Atmos. Chem. Phys.* **2015**, *15* (2), 845-865. DOI: 10.5194/acp-15-845-2015. Hennigan, C. J.; Miracolo, M. A.; Engelhart, G. J.; May, A. A.; Presto, A. A.; Lee, T.; Sullivan, A. P.; McMeeking, G. R.; Coe, H.; Wold, C. E.; et al. Chemical and physical transformations of organic aerosol from the photo-oxidation of open biomass burning emissions in an environmental chamber. *Atmos. Chem. Phys.* **2011**, *11* (15), 7669-7686. DOI: 10.5194/acp-11-7669-2011. Fleming, L. T.; Lin, P.; Roberts, J. M.; Selimovic, V.; Yokelson, R.; Laskin, J.; Laskin, A.; Nizkorodov, S. A. Molecular composition and photochemical lifetimes of brown carbon chromophores in biomass burning organic aerosol. *Atmos. Chem. Phys. Discuss.* **2019**, *2019*, 1-38. DOI: 10.5194/acp-2019-523. Mazzoleni, L. R.; Zielinska, B.; Moosmüller, H. Emissions of Levoglucosan, Methoxy Phenols, and Organic Acids from Prescribed

Burns, Laboratory Combustion of Wildland Fuels, and Residential Wood Combustion. *Environmental Science & Technology* **2007**, *41* (7), 2115-2122. DOI: 10.1021/es061702c.

(2) Sekimoto, K.; Koss, A. R.; Gilman, J. B.; Selimovic, V.; Coggon, M. M.; Zarzana, K. J.; Yuan, B.; Lerner, B. M.; Brown, S. S.; Warneke, C.; et al. High- and low-temperature pyrolysis profiles describe volatile organic compound emissions from western US wildfire fuels. *Atmos. Chem. Phys.* **2018**, *18* (13), 9263-9281. DOI: 10.5194/acp-18-9263-2018.

(3) Chen, Y.; Bond, T. C. Light absorption by organic carbon from wood combustion. *Atmos. Chem. Phys.* **2010**, *10* (4), 1773-1787. DOI: 10.5194/acp-10-1773-2010.

(4) Smith, D. M.; Fiddler, M. N.; Pokhrel, R. P.; Bililign, S. Laboratory studies of fresh and aged biomass burning aerosol emitted from east African biomass fuels – Part 1: Optical properties. *Atmos. Chem. Phys.* **2020**, *20* (17), 10149-10168. DOI: 10.5194/acp-20-10149-2020. Li, C.; He, Q.; Hettiyadura, A. P. S.; Käfer, U.; Shmul, G.; Meidan, D.; Zimmermann, R.; Brown, S. S.; George, C.; Laskin, A.; Rudich, Y. Formation of Secondary Brown Carbon in Biomass Burning Aerosol Proxies through NO3 Radical Reactions. *Environmental Science & Technology* **2020**, *54* (3), 1395-1405. DOI: 10.1021/acs.est.9b05641. Tóth, A.; Hoffer, A.; Nyirő-Kósa, I.; Pósfai, M.; Gelencsér, A. Atmospheric tar balls: aged primary droplets from biomass burning? *Atmos. Chem. Phys.* **2014**, *14* (13), 6669-6675. DOI: 10.5194/acp-14-6669-2014.

(5) Divisekara, T.; Schum, S.; Mazzoleni, L. Ultrahigh performance LC/FT-MS non-targeted screening for biomass burning organic aerosol with MZmine2 and MFAssignR. *Chemosphere* **2023**, *338*, 139403. DOI: https://doi.org/10.1016/j.chemosphere.2023.139403.

(6) Gao, P.; Deng, R.; Jia, S.; Li, Y.; Wang, X.; Xing, Q. Effects of combustion temperature on the optical properties of brown carbon from biomass burning. *J. Environ. Sci.* **2024**, *137*, 302-309. DOI: https://doi.org/10.1016/j.jes.2022.12.026.

---

## Author Comment (AC2)

We thank the Reviewer for the detailed and insightful comments and suggestions. The feedback provided, particularly regarding the LC-MS method optimization and semi-quantification approach, helped us enhance the scientific quality of the revised article. Our point-by-point responses to the specific comments, together with the list of changes made in the revised article, are provided below.

**Reviewer 2 comments:**

Overall, the manuscript is timely and provides an important comparison of multiple NTA workflows. This work will provide a benchmark for others to work towards when undertaking NTA and highlights the type of optimising and testing which should take place. The link with toxicity is a great step forward, especially highlighting the lack of correlation with general aerosol composition (O:C, H:C etc).

However, while I agree with the approach of NTA optimisation and semi-quantification, I have some major comments regarding the LC-MS and semi-quantification methods used. Overall, the manuscript should be published in ACP once comments are addressed.

**Specific comments:**

**Reviewer 2:** L125- What mobile phase (A or B) does the 5% apply to?

**Author's response:** The 5% and other percentage values apply to the mobile phase B component, organic solvent.

**Changes in the revised manuscript:** This paragraph was revised to clarify that the gradient elution program refers to the percentage of mobile phase component B in the mobile phase "The gradient elution program involved adjusting the amount of eluent B as follows:"

**Reviewer 2:** L135 – How were the methods optimised? Was it based on a mix of all 59 standards and/or spiked into the sample matrix?

**Author's response:** The methods were optimized using filter extracts to obtain quality MS/MS spectra by step-wise adjustment of CE at low, medium, and high levels in both ESI polarities. The threshold values in DDA methods were chosen to obtain approximately 4000 MS/MS spectra, based on previously published data reporting the number of unique molecules detected in similar samples. We agree that the optimization of MS and MS/MS conditions wasn't reported in sufficient detail.

**Changes in the revised manuscript:** An explanation about adjusting MS/MS conditions was added in Section 2.2: **"**After adjusting the DDA conditions using BrC filter extracts, a quality MS and MS/MS

spectra were obtained for approx. 4000 features, which is comparable with similar, combustion-related samples (Brege et al., 2021; Divisekara et al., 2023; Young et al., 2021; Graham et al., 2002).”

**Reviewer 2:** L137 – How were the 59 organic molecules determined? Based on previous literature and/or commercial availability? How representative of the overall composition are the standards? Are you biasing towards this functionality?

**Author's response:** The model compounds were selected to best mimic the molecular composition of biomass-related, water-soluble BrC based on the previously published data and availability of standards. Most of the model compounds are aromatics, functionalized (aromatic)acidic, phenols, or N-containing molecules. We note that our approach may be biased due to the analytical technique employed. Electrospray ionization (ESI) is more selective towards polar compounds, forming $[M+H]^+$ or $[M-H]^-$ ions. Providing further insights into the non-polar fraction of BrC (PAHs, alkanes, alkylated benzenes, etc.) would likely require the use of GC-EI/MS, which was beyond the scope of our work.

**Changes in the revised manuscript:** Table S1 was added in the SI, with references justifying the use of selected standards as (model)tracers of organic molecules detected in combustion-related emissions. This table is now referenced at the beginning of Section 2.3: “First, the workflows based on MS-FINDER, CFM-ID, Metaboanalyst, GNPS, and MZmine were tested using 59 model compounds. These molecules were selected to mimic the molecular composition of water-soluble BrC emitted by BB based on previously published results and standards availability (Tables S1 and S2), and included derivatives of cinnamic acid, nitrophenols, and polycarboxylic, furoic, and fatty acids.”

**Reviewer 2:** L155/6 – Not entirely sure what this means. How was the structure annotated just based on MS1?

**Author's response:** The annotation workflow is based on the MS-DIAL scoring approach, which combines MS1 similarity, MS2 similarity, retention time (RT) similarity, and isotope ratio similarity (Tsugawa et al., 2015)

$$Score\_MS-DIAL$$
$$= \frac{MS2\ similarity + \ MS1\ similarity + \ RT\ similarity\ + \ 0.5\ isotope\ ratio\ similarity}{3.5} \times 100$$

However, in our data, retention time and isotope ratio information were unavailable. Therefore, the scoring function was simplified to include only MS1 and MS2 similarities

$$Score\_this\ work = \frac{MS2\ similarity + \ MS1\ similarity}{2} \times 100$$

For cases with MS1-only data (no MS2 spectra acquired or no match found in MS2 library), structure suggestions were based solely on high-resolution mass matching (Tsugawa et al., 2015).

$$Score\_MS1\ only = MS1\ similarity = exp\left\{-0.5 \times \left(\frac{Mass_{sample} - Mass_{library}}{\delta_{mass}}\right)^2\right\}$$

*where $\delta_{mass}$ is mass tolerance parameter, set to 0.01 Da in this study.*

Thus, the software proposes candidate structures whose theoretical exact masses and isotope distribution profiles in databases match the observed MS1 spectra within the specified tolerance. However, without MS/MS fragmentation data or retention time information, such matches are inherently ambiguous, reflecting lower confidence. "Suggested structures" describe a "grey zone", where evidence exists for possible structures, but insufficient information is available to achieve a higher confidence level (Schymanski et al., 2014).

**Changes in the revised manuscript:** To improve clarity, we have included these mathematical formulations and a discussion of the MS1-only annotation method in the SI (a new Section S3.1.5 was added), and it is now referenced in the main text: "The scoring approach implemented in MS-DIAL is described in more detail in Section S3.1.5."

**Reviewer 2:** Section 2.5, You have run 59 surrogate standards for library building/identification. When authentic standards were identified (level 1) in your BB samples, did you use the authentic standard for quantification? If so, why not? Could you have assessed the error of the semi-quantification technique based upon comparisons between authentic and surrogate standard quantification?

**Author's response:** The quantitative results were further analyzed to assess the difference between quantification results using authentic and surrogate standards. Furthermore, additional calibration was performed for model compounds detected in water-extractable BrC. The quantification biases ranged from -50 to 44% with an average -14 %, consistent with previously published data and with 50% uncertainty assumed for all concentrations derived using surrogate standards, as described in Section 2.5 of the main text.

| No. | Ion mode | Compound detected in BrC | Concentration (mg/L) | | Bias $= (\frac{C1 - C2}{C1}, \%)$ |
|-----|----------|--------------------------|----------------------|----|----|
| | | | Authentic (C1) | Surrogate (C2) | |
| 1 | | Glutaric acid | 5.0 | 7.4 | 44 |
| 2 | | Coumaric acid | 15.1 | 18.9 | 25 |
| 3 | (-) | Benzoic acid | 2.2 | 1.1 | -50 |
| 4 | (-) | Azelaic acid | 12.7 | 10.5 | -17 |
| 5 | | Suberic acid | 4.1 | 2.1 | -49 |
| 6 | | Homovanillic acid | 3.4 | 3.1 | -8.8 |

| No. | Ion mode | Compound detected in BrC | Concentration (mg/L) | | Bias $= (\dfrac{C1 - C2}{C1}, \%)$ |
| --- | --- | --- | --- | --- | --- |
| | | | Authentic (C1) | Surrogate (C2) | |
| 7 | | Orcinol | 3.3 | 4.0 | 21 |
| 8 | | Syringic acid | 3.5 | 1.8 | -49 |
| 9 | (+) | Cinnamic acid | 3.4 | 2.1 | -38 |
| 10 | | Syringaldehyde | 18.6 | 15.4 | -17 |

At the same time, we believe that providing further insights into the results of semi-quantification (for instance for all surrogate standards) would require a separate, dedicated investigation, as this is a large, complex and dynamically evolving area (Evans et al., 2024; Malm et al., 2021; Kruve, 2019).

**Changes in the revised manuscript:** A new section was added in the SI, "S2. Semi-quantification using surrogate standards". The above Table (now Table S4) was added in the SI and is now referenced in Section S2 "This uncertainty is consistent with the quantitative data obtained using authentic and surrogate standards for the compounds detected in water-soluble BrC (Table S4) and higher than the effects of the sample matrix (Table S5)."

**Reviewer 2:** How were the 5 surrogate standards chosen?

**Author's response:** The five surrogate standards (for each ionization mode) were chosen to cover the range of retention times of water-soluble tracers detected in BrC. We prioritized the molecules with good chromatographic properties under the LC/MS analysis conditions used. Initially, a larger number of candidates were tested before narrowing the list to the 5 standards.

**Changes in the revised manuscript:** A justification for selecting the specific surrogate standards was added in the SI, before Table S3 "The five surrogate standards (for each ionization mode) were chosen to cover the range of retention times of water-soluble organics in BrC. Initial tests were performed on a larger number of molecules, and the list was narrowed down, prioritizing the standards with good chromatographic properties under the LC/MS analysis conditions used – Table S3".

**Reviewer 2:** Why does tetradodecanoic acid cover the tr range of above 35 minutes with a retention time of 26 minutes?

**Author's response:** This is a typing error.

**Changes in the revised manuscript:** The $t_r$ for tetradecanedioic acid in Table S3 (formerly Table S2) was corrected to 39.33 min.

**Reviewer 2:** Why weren't all 59 standards used for semi-quantification? Your range used by each standard is large; including more standards would have narrowed the windows.

**Author's response:** We agree that using more standards could yield more accurate results. At the same time, large uncertainties are an inherent feature of semi-quantification with surrogate standards. As such, we believe that in the context of our work, using more surrogate standards wasn't critical, as we wanted to obtain an estimate of the amount of water-soluble BrC using the LC/MS data. Even if the quantification results from the three methods agree very well (as presented in Fig. 7 in the main text), the LC/MS data is still implied to a large uncertainty, as underlined in Section 3.3.

**Changes in the revised manuscript:** Description in Section 2.5 was revised as "Five surrogate standards were used in each ionization mode, with retention times from 2 to 40 min (Table S3), covering the elution window for WSOCs detected in BrC$_{aq}$." to underline that the retention times of surrogate standards used converted the retention times of the investigated analytes.

**Reviewer 2:** Did you spike the standards into the sample matrix to determine matrix effects on concentrations, given the sample complexity?

**Author's response:** Additional analyses were performed to analyze the matrix effects, particularly the impact on ionization efficiency and detection sensitivity (Williams et al., 2023). Please note also that the filter extracts were diluted 10 times before LC/MS analyses to reduce matrix effects (Pihlström et al., 2021) and avoid instrument contamination.

The matrix effect for each analyte was evaluated by comparing the analyte response (peak area) in the diluted sample extract with that in a pure solvent standard, and calculated using the following equation

$$\% \, Matrix \, Effect = \frac{(Peak \, area_{sample} - Peak \, area_{solvent})}{Peak \, area_{solvent}} * 100$$

| Concentration (mg/L) | Negative ion mode | | | Positive ion mode | | | |
|---|---|---|---|---|---|---|---|
| | Adipic acid | Caffeic acid | Sebacic acid | Trytophan | Isatin | Fenuron | Isoproturon |
| 1 | 19 | 10 | 9.2 | -20 | -40 | -20 | -8.6 |
| 2 | 8.4 | 2.4 | 3.9 | -13 | -32 | -19 | -7.5 |
| 5 | 1.4 | 6.9 | 4.6 | -8.2 | -27 | -20 | -9.2 |
| 10 | 6.9 | 3.4 | 3.5 | -8.8 | -22 | -14 | -8.4 |
| 20 | 2.0 | 9.3 | 8.4 | -11 | -13 | -11 | -7.2 |

The above Table summarizes the matrix effect values for surrogate compounds measured in both negative and positive ionization modes across different concentration levels (1-20 mg/L). The results

indicate that the matrix effect for all compounds remained below 20%, except isatin, which exhibited higher suppression ranging from -13% to -40%. According to the criteria defined in SANTE/11312/2021v2, these results indicate that matrix effects were generally within acceptable limits for most analytes in this workflow (Pihlström et al., 2021). Furthermore, matrix effects were lower than the uncertainty resulting from the use of surrogate standards.

**Changes in the revised manuscript:** The above discussion, Table S5 were added in a new Section S2. "Semi-quantification using surrogate standards" in the SI.

Line was added in Section 2.1: "The extracts were diluted tenfold before LC/MS analysis to prevent ion source contamination and minimize matrix effects on the semi-quantitative results (see Section 2.5)."

Lines were added in Section 2.5, regarding the use of surrogate standards and the results of semi-quantitative analyses: "For compounds detected in $BrC_{aq}$, this value is consistent with the results obtained using authentic and surrogate standards (Table S4). Furthermore, the imposed 50% uncertainty exceeds the matrix effects observed for most surrogate standards (Table S5). To prevent introducing a positive bias into the data, molecules detected in both ionization modes were identified by comparing annotation results, and their average concentrations were used to obtain the amount of $BrC_{aq}$."

**Reviewer 2:** Section 3- Many of the analytes will ionise in both the negative and positive mode. How were these species dealt with in terms of final concentrations and composition?

**Author's response:** We agree that detecting the same molecules in both modes would lead to a positive bias in the semi-quantification results. The molecules detected in both modes were identified by comparing the annotation results, and the average concentrations were used for such compounds.

**Changes in the revised manuscript:** A clarification was added in Section 2.5. "To prevent introducing a positive bias into the quantitative data, molecules detected in both ionization modes were identified by comparing annotation results, and their average concentrations were used to obtain the amount of $BrC_{aq}$."

**Reviewer 2:** L235 – What sample did you use to determine this? I.e a mixture of standards/spiked sample

**Author's response:** The workflows were assessed using a standard mixture of 59 model compounds.

**Changes in the revised manuscript:** The caption of Figure 5 was revised as "Workflow performance evaluation with the standard mixture of 59 model compounds (Table S1). Number of correctly identified analytes (A) and number of analytes identified only by a given software (B)"

**Reviewer 2:** L239 The use of "identifies" here makes it seem like this is a known compound, maybe change to annotated?

**Author's response:** We agree that the term "annotated" is more appropriate in the context of NTA.

**Changes in the revised manuscript:** This sentence was revised as "The combination of MS-DIAL and MS-FINDER also annotated 5 unique features not recognized by any other workflow, whereas MZmine annotated only one unique analyte – Fig. 5."

**Reviewer 2:** Fig 6 – How representative are your standards to the overall detected composition?

**Author's response:** Naturally, it is challenging to cover the molecular diversity of BrC using commercially available standards. At the same time, the values of Kendrick Mass and Kendrick Mass Defect, double-bond equivalent, O/C and H/C ratios, C+N, and C numbers for the selected standards are all within the range of the values presented in Fig. 6 for the main components (represented by the largest cycles) of water-extractable BrC.

**Changes in the revised manuscript:** Table S2 was added in the SI, reporting the values of the parameters presented in Fig. 6 and elemental composition of all 59 model compounds, including an average and two standard deviation values for each parameter.

A paragraph was added in the SI, before Table S2: **"**The general characteristics of surrogate standards used to evaluate the workflows were derived as described in Section S4 and are listed in Table S2. Covering the high molecular diversity of the water-extractable BrC with commercially available standards is challenging. At the same time, values of Kendrick Mass and Kendrick Mass Defect, double-bond equivalent, O/C and H/C ratios, C+N, and C numbers for the selected standards are all within the range of the values presented in Fig. 6 in the main text, for the main components of water-extractable BrC.**"**

Line was added in the caption of Fig. 6 in the main text: **"**These general characteristics (Section S4) of water-extractable BrC were comparable to those of the surrogate standards used to evaluate the NTA workflows (Table S2).**"**

Table S2 is now referenced in Section 2.3.

**Cited references**

Brege, M. A., China, S., Schum, S., Zelenyuk, A., and Mazzoleni, L. R.: Extreme Molecular Complexity Resulting in a Continuum of Carbonaceous Species in Biomass Burning Tar Balls from Wildfire Smoke, ACS Earth and Space Chemistry, 5, 2729-2739, 10.1021/acsearthspacechem.1c00141, 2021.

Divisekara, T., Schum, S., and Mazzoleni, L.: Ultrahigh performance LC/FT-MS non-targeted screening for biomass burning organic aerosol with MZmine2 and MFAssignR, Chemosphere, 338, 139403, https://doi.org/10.1016/j.chemosphere.2023.139403, 2023.

Evans, R. L., Bryant, D. J., Voliotis, A., Hu, D., Wu, H., Syafira, S. A., Oghama, O. E., McFiggans, G., Hamilton, J. F., and Rickard, A. R.: A Semi-Quantitative Approach to

Nontarget Compositional Analysis of Complex Samples, Anal. Chem., 96, 18349-18358, 10.1021/acs.analchem.4c00819, 2024.

Graham, B., Mayol-Bracero, O. L., Guyon, P., Roberts, G. C., Decesari, S., Facchini, M. C., Artaxo, P., Maenhaut, W., Köll, P., and Andreae, M. O.: Water-soluble organic compounds in biomass burning aerosols over Amazonia 1. Characterization by NMR and GC-MS, Journal of Geophysical Research: Atmospheres, 107, LBA 14-11-LBA 14-16, https://doi.org/10.1029/2001JD000336, 2002.

Kruve, A.: Semi-quantitative non-target analysis of water with liquid chromatography/high-resolution mass spectrometry: How far are we?, Rapid Commun Mass Spectrom, 33, 54-63, 2019.

Malm, L., Palm, E., Souihi, A., Plassmann, M., Liigand, J., and Kruve, A.: Guide to Semi-Quantitative Non-Targeted Screening Using LC/ESI/HRMS, Molecules, 26, 10.3390/molecules26123524, 2021.

Pihlström, T., Fernández-Alba, A. R., Amate, C. F., Poulsen, M. E., Hardebusch, B., Anastassiades, M., Lippold, R., Cabrera, L. C., de Kok, A., and ORegan, F.: Analytical quality control and method validation procedures for pesticide residues analysis in food and feed SANTE 11312/2021, Sante, 11312, 2021.

Schymanski, E. L., Jeon, J., Gulde, R., Fenner, K., Ruff, M., Singer, H. P., and Hollender, J.: Identifying Small Molecules via High Resolution Mass Spectrometry: Communicating Confidence, Environmental Science & Technology, 48, 2097-2098, 10.1021/es5002105, 2014.

Tsugawa, H., Cajka, T., Kind, T., Ma, Y., Higgins, B., Ikeda, K., Kanazawa, M., VanderGheynst, J., Fiehn, O., and Arita, M.: MS-DIAL: data-independent MS/MS deconvolution for comprehensive metabolome analysis, Nat Methods, 12, 523-526, 10.1038/nmeth.3393, 2015.

Williams, M. L., Olomukoro, A. A., Emmons, R. V., Godage, N. H., and Gionfriddo, E.: Matrix effects demystified: Strategies for resolving challenges in analytical separations of complex samples, Journal of Separation Science, 46, 2300571, 2023.

Young, T. M., Black, G. P., Wong, L., Bloszies, C. S., Fiehn, O., He, G., Denison, M. S., Vogel, C. F. A., and Durbin-Johnson, B.: Identifying Toxicologically Significant Compounds in Urban Wildfire Ash Using In Vitro Bioassays and High-Resolution Mass Spectrometry, Environmental Science & Technology, 55, 3657-3667, 10.1021/acs.est.0c06712, 2021.